# The Sample Complexity of Online Strategic Decision Making with Information Asymmetry and Knowledge Transportability

**Jiachen Hu**[1]  **Rui Ai**[2]  **Han Zhong**[3]  **Xiaoyu Chen**[4]  **Liwei Wang**[3 4]  **Zhaoran Wang**[5]  **Zhuoran Yang**[6]

## Abstract

Information asymmetry is a pervasive feature of multi-agent systems, especially evident in economics and social sciences. In these settings, agents tailor their actions based on private information to maximize their rewards. These strategic behaviors often introduce complexities due to confounding variables. Simultaneously, knowledge transportability poses another significant challenge, arising from the difficulties of conducting experiments in target environments. It requires transferring knowledge from environments where empirical data is more readily available. Against these backdrops, this paper explores a fundamental question in online learning: Can we employ non-i.i.d. actions to learn about confounders even when requiring knowledge transfer? We present a sample-efficient algorithm designed to accurately identify system dynamics under information asymmetry and to navigate the challenges of knowledge transfer effectively in reinforcement learning, framed within an online strategic interaction model. Our method provably achieves learning of an $\epsilon$-optimal policy with a tight sample complexity of $\tilde{O}(1/\epsilon^2)$.

## 1. Introduction

Multi-agent systems are widely applied in reinforcement learning (RL) (Littman, 1994; Wang et al., 2019; Dubey & Pentland, 2021), economics (Brero et al., 2022), social science (Sabater & Sierra, 2005), and robotics (Yan et al., 2013). In systems like these, agents are characterized by their diverse private information and their drive to maximize individual rewards, which give rise to what is termed information asymmetry (Myerson, 1982; Gan et al., 2022a). Due to information asymmetry, the agents always strategically choose their actions according to the predetermined and publicly known policy of the principal, while keeping their private information concealed from the principal's awareness. Here, we say agents are "strategical" because they will choose their strategic actions based on personal types and aim at maximum obtainment. A famous example in economics is the generalized principal-agent problem (Myerson, 1982), where a principal interacts with several myopic agents with private types and type-based strategic bidding.

On the other hand, it's also common and sometimes necessary for the learner in machine learning to generalize the knowledge from one domain to other related domains, which is often referred to as transfer learning (Zhuang et al., 2020). Similar phenomena have also been observed in casual inference, which is known as knowledge transportability (Pearl & Bareinboim, 2011). Knowledge transportability studies the problem of transferring information learned from experiments to different environments, where conducting active experiments is difficult or unfeasible but only passive observations are allowed. Social problems, particularly medical issues, frequently involve instances of knowledge transportability (Pearl & Bareinboim, 2011; Bareinboim & Pearl, 2013). A motivating example is to transfer the learned effect of a treatment from one population (e.g., experiments done in New York City) with a large number of samples to other populations where massive experiments might be not available (e.g., investigating the outcome of that treatment in Los Angeles).

In this paper, we hope to theoretically study online decision-making under information asymmetry arising from the agents' side and knowledge transportability needed by the principal. Inspired by the generalized principal-agent problem, we model a scenario in which a principal sequentially interacts with a series of agents. Each agent, driven by short-term objectives, myopically maximizes her rewards based on her private information (Zhong et al., 2021). This model captures the dynamics of a principal engaging with a con-

---

[1]School of Computer Science, Peking University [2]Institute for Data, Systems, and Society, Massachusetts Institute of Technology [3]Center for Data Science, Peking University [4]National Key Laboratory of General Artificial Intelligence, School of Intelligence Science and Technology, Peking University [5]Northwestern University [6]Yale University. Correspondence to: Zhuoran Yang <zhuoran.yang@yale.edu>.

*Proceedings of the 42nd International Conference on Machine Learning*, Vancouver, Canada. PMLR 267, 2025. Copyright 2025 by the author(s).

tinuous flow of agents where their private types and actions directly affect the transition and the reward of the principal. The goal of the principal is to design a policy to maximize his total rewards when interacting prospectively with a specific target population of agents, though the online data may come from a different population of agents thus knowledge transfer is necessary. As a result, we formalize this problem as an online strategic interaction model–a generalization of the strategic Markov decision process (MDP) similar to Yu et al. (2022), which studies an offline setting. However, we must adaptively adjust our policy, i.e., time-varying, in an online learning setting to explore unknown environments. Therefore, our actions are not i.i.d. distributed, but are interdependent. Hence, the traditional concentration inequalities used in Yu et al. (2022) do not apply to our case. This immediately raises a question:

*Is it possible to learn a model with confounders using non-i.i.d. actions and states?*

There are two main challenges in our model in order to learn the optimal policy of the principal. The first one is the presence of unobserved confounders–the agents' private types and private actions, which directly affect both the rewards and transition dynamics but are kept unobserved to the principal. The second is to transfer the learned information from online data to the target population of agents. It prompts a second question:

*Can we design an approximately optimal algorithm if the target distribution and source distribution are different, and how does this difference affect the sample complexity?*

Different from standard RL, the exploration needed here is more challenging because the heterogeneous agents can strategically manipulate their feedback to the principal, which affects the rewards and transitions.

We provide affirmative answers to both questions. To resolve the challenges, we propose a model-based algorithm to learn a nearly optimal policy of the principal. Our algorithms leverage a novel nonparametric instrumental variable (NPIV) method sparked by Angrist & Imbens (1995); Angrist & Krueger (2001); Newey & Powell (2003); Ai & Chen (2003) with algorithmic instruments to establish causal identification of the system in the presence of confounders. With such identification, we're able to construct a high probability confidence set for the model or the value functions under the target domain to transfer knowledge (Pan & Yang, 2009; Taylor & Stone, 2009) using the method of moments. Our analysis shows that such a model-based algorithm provably learns a $\epsilon$-optimal policy with only $\tilde{O}(1/\epsilon^2)$ samples which matches corresponding lower bounds up to logarithmic terms.

In summary, our contributions are two-fold:

- In order to theoretically study reinforcement learning with information asymmetry and knowledge transportability, we introduce the online strategic interaction model motivated by the strategic MDP (Yu et al., 2022). Our online strategic interaction model breaks the i.i.d. data condition and presents a scenario in which reinforcement learning models can be learned using time-varying data. Technically, we propose a model-based algorithm in the general MDP setting which leverages the NPIV method to do causal identification of the underlying model.

- We investigate the conditions under which our algorithm can provably learn a near-optimal policy. Using $\tilde{O}(1/\epsilon^2)$ samples, the proposed model-based algorithm learns an $\epsilon$-optimal policy and we show its dependency on the knowledge transportability. We define a knowledge transfer multiplicative term $C^{\mathrm{f}}$ and use it as a measure of the hardness increased by transferring knowledge (cf. Section 5).

### 1.1. Related Work

Our work is closely connected to several bodies of literature discussed below.

**RL in economics.** Many models or problems extensively studied in economics have been combined with reinforcement learning, including Stackelberg game (Başar & Olsder, 1998; Zhong et al., 2021), Bayesian persuasion (Kamenica & Gentzkow, 2011; Gan et al., 2022b; Wu et al., 2022), mechanism design (Gan et al., 2022a; Bernasconi et al., 2022), performative prediction (Mandal et al., 2022), etc. Different from those models, our work is a natural extension of the generalized principal-agent problem to reinforcement learning focusing on learning the optimal policy of the principal in the presence of information asymmetry and knowledge transportability.

**Efficient exploration in RL.** The exploration problem has been extensively studied in tabular MDPs (Auer et al., 2008; Azar et al., 2017; Dann et al., 2017; Jin et al., 2018; Dann et al., 2019; Zanette & Brunskill, 2019; Zhang et al., 2022), MDPs with linear function approximation (Yang & Wang, 2019; Jin et al., 2020; Yang & Wang, 2020; Cai et al., 2020; Ayoub et al., 2020; Zhou et al., 2021; Hu et al., 2021; Chen et al., 2021) and general function approximation (Russo & Van Roy, 2013; Jiang et al., 2017; Sun et al., 2019; Jin et al., 2021; Du et al., 2021). Apart from standard RL models, the strategic interaction model poses additional challenges due to, for instance, the strategic behaviors of agents caused by information asymmetry. Since the principal cannot observe agents' types, partially observed feedbacks bring higher uncertainty and more nuisance in algorithm design.

**RL with confounded Data.** There is a line of works studying reinforcement learning in the presence of confounded data (Chen & Zhang, 2021; Wang et al., 2021; Liao et al., 2021; Shi et al., 2021; Bennett et al., 2021; Bennett & Kallus, 2021; Wang et al., 2022; Lu et al., 2022; Yu et al., 2022). The studies of Chen & Zhang (2021); Liao et al. (2021); Yu et al. (2022) also use the instrumental variables to do causal identification. Among them, Yu et al. (2022) is most related to this work, which studied strategic MDPs in the offline setting. In contrast, we study the online strategic interaction model together with knowledge transfer, a generalization of their model with a break of i.i.d. assumption, and distribution shift. We provide an elaborate summary of the novelty in this work beyond Yu et al. (2022) and standard RL in Appendix B.2.

**The principal-agent problem and instrumental variable model.** The principal-agent problem is well-known in economics (Myerson, 1982; Guruganesh et al., 2021; Zhang & Conitzer, 2021; Gan et al., 2022a), which features the strategic interactions between a principal and an agent with private type and private action. The challenges in the principal-agent problem are known as "moral hazard" (incomplete information about actions) and "adverse selection" (incomplete information about types), which are both present in our work. The instrumental variable model has been extensively studied in economics (Angrist & Imbens, 1995; Angrist & Krueger, 2001; Ai & Chen, 2003; Newey & Powell, 2003; Blundell et al., 2007; Chen & Pouzo, 2012; Chen & Qi, 2022), with applications in (statistical) machine learning (Harris et al., 2022) and reinforcement learning (Chen & Zhang, 2021; Liao et al., 2021; Yu et al., 2022). As an application of the instrumental variable model, we use the non-parametric instrumental variable model to build conditional moment equations and methods of moments to identify the underlying model.

## 2. Preliminaries

We use Markov decision processes to depict the standard online interaction models involved in this paper, which can be summarized as $(\mathcal{S}, \mathcal{A}, P, R, H, s_1)$. $\mathcal{S}$ and $\mathcal{A}$ denote the state space and action space, respectively. In the meanwhile, $P = \{P_h(\cdot \mid s, a)\}_{h=1}^H$ and $R = \{R_h(s, a)\}_{h=1}^H$ are the transition dynamics and reward functions, which are both unknown. $H$ is the time horizon while $s_1$ denotes the initial state.[1]

For each episode, a player interacts with the model starting from state $s_1$ for $H$ steps. When the player reaches state $s_h$, he receives an observation indicating $s_h$. So, without loss of

generality, we assume he can observe $s_h$ directly. Then he takes an action $a_h$ according to the past states and actions, receives a reward $r_h = R_h(s_h, a_h)$ and transits to the next state $s_{h+1} \sim P_h(\cdot \mid s_h, a_h)$.

We consider the time-inhomogeneous Markov policy class $\Pi$ in this work. A policy $\pi = \{\pi_h\}_{h=1}^H \in \Pi$ maps each $s_h$ to $\Delta(\mathcal{A})$ at step $h$. The value function $V_h^\pi : \mathcal{S} \to \mathbb{R}$ of a policy $\pi$ at step $h$ conditioned on $s_h$ is defined as

$$V_h^\pi(s) \stackrel{\text{def}}{=} \mathbb{E}_\pi\Big[\sum_{t=h}^H r_t \mid s_h = s\Big].$$

The goal is to learn an optimal policy $\pi^*$ with a highest cumulative rewards, *viz*,

$$\pi^* = \arg\max_{\pi \in \Pi} V_1^\pi(s_1).$$

## 3. The Online Strategic Interaction Model

The online strategic interaction model formalizes the sequential strategic interactions of a principal and $H$ agents, where each agent possesses a different private type sampled from a prior distribution.[2] To elaborate further, at each step $h \in [H] = \{1, ..., H\}$, an agent with private type $t_h$ arrives and chooses a strategic action $b_h$ that maximizes her own payoff. The principal needs to dynamically adjust his strategy to actively engage in exploration, with the goal of learning an approximately optimal policy as quickly as possible. This prior distribution represents the population from which the online data is collected, which may be different from the target population of agents, underscoring the necessity of knowledge transfer. We also call the prior distribution as the source distribution $\mathcal{P}^s = \{\mathcal{P}_h^s\}_{h=1}^H$. Starting at a fixed state $s_1$, the interactions happen as follows (see Figure 1 for a more intuitive understanding):

- For the $h$-th agent, the principal transits to state $s_h$ and then takes an action $a_h$ according to his strategy.

- The agent's private type $t_h$ is sampled from the unknown source population $\mathcal{P}_h^s$. Based on it, the myopic agent strategically takes an action $b_h = \arg\max_b R_h^a(s_h, a_h, t_h, b)$ to maximize her own reward $R_h^a$. Note that $t_h$ and $b_h$ are both unobserved by the principal (Maskin & Riley, 1984).

- After the agent takes her action, the principal receives a manipulated feedback $e_h \sim \tilde{F}_h(\cdot \mid s_h, a_h, t_h, b_h)$ from the agent according to her private information. Denote $F_h(\cdot \mid s_h, a_h, t_h) \stackrel{\text{def}}{=} \tilde{F}_h(\cdot \mid s_h, a_h, t_h, \arg\max_b R_h^a(s_h, a_h, t_h, b))$, then $e_h \sim F_h(\cdot \mid s_h, a_h, t_h)$.

---

[1] We assume the initial state is a fixed state for simplicity. It's straightforward to extend it to the case where $s_1$ is sampled from a fixed initial distribution.

[2] It can also model the interaction of the principal and a single agent for $H$ steps, who has a different aspect of personal type drawn from a prior distribution at each step.

- Finally, the principal receives a reward $r_h = R_h^*(s_h, a_h, e_h) + \xi_h$, where $R_h^*$ is an unknown function and $\xi_h$ is assumed to be an unobserved endogenous zero-mean noise (i.e., confounded with the private type $t_h$ referring to Appendix C). The principal transits to the next state $s_{h+1} \sim P_h^*(\cdot \mid s_h, a_h, e_h)$ according to an unknown function $P_h^*$. Similarly, we could also allow the stochastic transition of $s_{h+1}$ to be endogenous or confounded with $t_h$. Then the principal starts the interaction with the next agent.

### 3.1. Motivation

Next, we discuss the motivation for designing the online strategic interaction model and use two real-world examples of the existence of confounders to digest this setting. More details can be found in Appendix C.

As mentioned in the introduction, the main motivation to study the online strategic interaction model is to understand the strategy design of multi-agent strategic interaction (e.g., the generalized principal-agent model) in a sequential decision-making setting, where both problems are common in reality. To better align this model with the real world, we incorporate several generalizations including the **distribution shift** of the agent population, the **extensively large state space**, and the **endogenous noise** in our model. Each of these generalizations greatly enhances the expressive power of our model in the sense that

- Without distribution shift (i.e., the knowledge transportability), the model reduces to single-agent RL, which is somehow not hard to compute near-optimal policies. The distribution shift condition initiates many challenges studied in the paper such as the confounding issue and the knowledge transfer.

- A large state space is ubiquitous in complex real-world problems. Our analysis characterizes the impact of problem size growth on the sample complexity of our algorithms, for instance, through the Eluder dimension.

- Many practical applications have endogenous noises while existing work only assumes exogenous noise. In real-world problems, the principal is typically a company or an organization, and the agents are clients, employees, etc. The state $s_h$ is usually the conditions of the company, and the rewards are the profits of the company over an agent, which has a deep connection to the private type of the agents (e.g., the personality, the health conditions). Thus, the distribution of the rewards is largely correlated to the private type. If the noises are exogenous variables, this correlation will be realized solely by the feedback $e_h$. That is to say, the private type must influence the reward distribution

via an observable term $e_h$. However, the feedback $e_h$ is observed so that it only conveys very limited information of the private type. In reality, the noise term $\xi_h$ may include the unobserved confounder in the reward that is not measured even when intervening with the feedback $e_h$. Please see the next section for some examples for further explanations.

*Example* 3.1 (Contract design). The shareholders of a company aim to maximize the stock price $s_h$ and related returns. At each step $h$, the company decides whether to replace the CEO and what her compensation should be denoted by $a_h$. CEOs can be either diligent or lazy, namely, $t_h \in \{\text{Diligent}, \text{Lazy}\}$. Different types of CEOs will choose to exert different levels of effort $b_h$, and whether a CEO is hardworking affects some hidden factors, e.g., the morale of the company's employees. Shareholders cannot observe the CEO's effort directly; they can only observe the operational status of the company $e_h$. At time $h$, the shareholders' returns are influenced by several factors: the stock price and the operational status of the company which determine dividends, the CEO's salary, and employee morale (which may affect turnover rates). The first two terms constitute $R_h^*(s_h, a_h, e_h)$ while the last unobserved term forms $\xi_h$. This explains the source of confounders and ends up with reward $r_h = R_h^*(s_h, a_h, e_h) + \xi_h$. The market can observe the company's stock price, financial statements, and operational condition at moment $h$, and these observations influence the stock price at $h+1$, saying $s_{h+1} \sim P_h^*(\cdot \mid s_h, a_h, e_h)$ through self-fulfilling expectation (Hamilton & Whiteman, 1985).

Intuitively, this model enables agents to strategically manipulate rewards of the principal and state transitions via their private information through their feedback $e_h$, which is more challenging than classic RL. The introduction of endogenous variables greatly enhances the generality of our model but also raises more challenges to designing efficient algorithms. We provide more details of the model and the motivation, as well as more motivating examples in Appendix C. We use $\mathcal{M}^*(\mathcal{P}^{\mathrm{s}})$ to denote the online strategic interaction model under the source distribution and let $\mathcal{P}^{\mathrm{t}} = \{\mathcal{P}_h^{\mathrm{t}}\}_{h=1}^H$ be the target population of agents. We close off this section by providing an application scenario of knowledge transfer.

*Example* 3.2 (Experimental design.). With the rapid development of Large Language Models (LLMs), integrating LLMs into experimental design (Kumar et al., 2023), such as A/B testing, has become a new trend. Compared to the high costs associated with using human samples for experiments, the cost of using LLMs is essentially zero. However, LLMs exhibit significant differences from humans in many aspects. For example, in addressing an optimization problem, LLMs operate with complete rationality, whereas humans sometimes possess only bounded rationality (Simon, 1990;

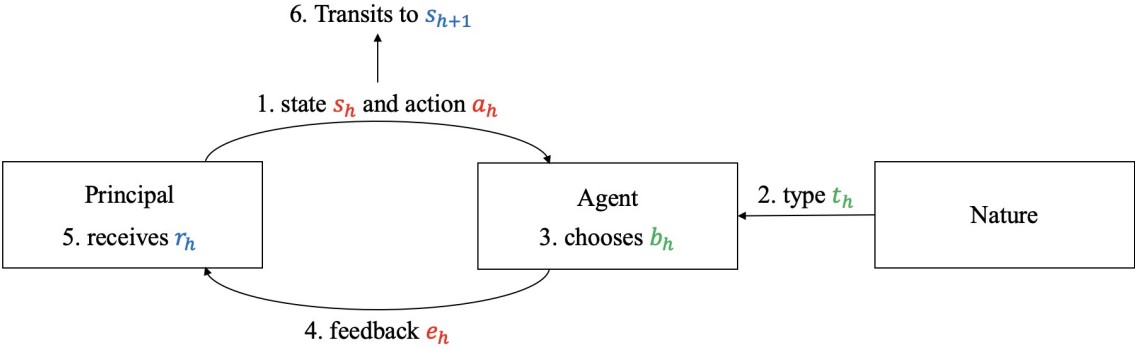

*Figure 1.* Timeline of the interaction. $r_h$ and $s_{h+1}$ (blue) are influenced by observable $s_h, a_h, e_h$ (red) and unobservable $t_h$ (green). We use numbers to indicate the sequence of events.

Conlisk, 1996). Therefore, experiment designers need to combine known human characteristics say $\mathcal{P}^t$ with experimental data derived from LLM features say $\mathcal{P}^s$ to develop optimal mechanisms tailored for human agents.

### 3.2. Planning in the Online Strategic Interaction Model

Since we are trying to design model-based algorithms, we introduce the planning algorithm in the online strategic interaction model in this section. If we are given the underlying rewards $\{R_h^*\}_{h=1}^H$ and transitions $\{P_h^*\}_{h=1}^H$, we can construct an aggregated model with respect to a target type distribution $\mathcal{P}^t$, i.e., assuming the random type is sampled from $\mathcal{P}^t$.

Define the aggregated model $\bar{\mathcal{M}}^* = (\mathcal{S}, \mathcal{A}, \bar{R}^*, \bar{P}^*, H, s_1)$ under the target distribution as

$$\bar{R}_h^*(s_h, a_h) \stackrel{\text{def}}{=} \mathbb{E}_{t \sim \mathcal{P}_h^t, e \sim F_h(\cdot|s_h, a_h, t)} \left[ R_h^*(s_h, a_h, e) \right],$$
(3.1)

$$\bar{P}_h^*(\cdot \mid s_h, a_h) \stackrel{\text{def}}{=} \mathbb{E}_{t \sim \mathcal{P}_h^t, e \sim F_h(\cdot|s_h, a_h, t)} \left[ P_h^*(\cdot \mid s_h, a_h, e) \right].$$
(3.2)

It is known that the reward and transition dynamics in the target model $\mathcal{M}^*(\mathcal{P}^t)$ is equivalent to $\bar{\mathcal{M}}^*$. We provide a brief explanation here and defer the details to Appendix C.

Given any $(s_h, a_h, e_h)$ in $\mathcal{M}^*(\mathcal{P}^t)$, we know $\xi_h$ is an endogenous noise which may be confounded with $t_h$. This means

$$\mathbb{E}\left[\xi_h \mid s_h, a_h, e_h\right] \neq 0,$$
(3.3)

since $e_h$ also correlates with $t_h$. Nevertheless, we note that $\xi_h$ is independent of $(s_h, a_h)$ in our model as apart from what absorbed in $R_h^*(s_h, a_h, e_h)$, the rest only depends on the unobservable type $t_h$, which implies that

$$\mathbb{E}_{t_h \sim \mathcal{P}_h^t, e_h \sim F_h(\cdot|s_h, a_h, t_h)} \left[\xi_h \mid s_h, a_h\right] = 0,$$

because $\xi_h$ is a zero-mean noise. Therefore, we conclude that

$$\mathbb{E}_{\mathcal{M}^*(\mathcal{P}^t)} \left[r_h \mid s_h, a_h\right] \stackrel{\text{def}}{=} \mathbb{E}_{t_h \sim \mathcal{P}_h^t, e_h \sim F_h(\cdot|s_h, a_h, t_h)} \left[r_h \mid s_h, a_h\right]$$
$$= \bar{R}_h^*(s_h, a_h),$$

and analogously the transition of $\mathcal{M}^*(\mathcal{P}^t)$ is also identical to $\bar{P}_h^*(\cdot \mid s_h, a_h)$.

Now it suffices to learn the optimal policy $\bar{\pi}^*$ of $\bar{\mathcal{M}}^*$. Since the aggregated model is an episodic MDP, its optimal policy is a Markov policy, which justifies the previous definition of $\Pi$. Denote the value function of any policy $\pi$ on $\bar{\mathcal{M}}^*$ as $\bar{V}_{\bar{\mathcal{M}}^*}^\pi$, we say a policy $\pi$ is $\epsilon$-optimal if

$$\bar{V}_{\bar{\mathcal{M}}^*}^\pi \geq \bar{V}_{\bar{\mathcal{M}}^*}^{\bar{\pi}^*} - \epsilon.$$

### 3.3. Notation Guide

We use $d_h^{s,\pi}(\cdot, \cdot, \cdot)$ (resp. $d_h^{t,\pi}(\cdot, \cdot, \cdot)$) to denote the joint distribution of $s_h, a_h, e_h$ for policy $\pi$ under model $\mathcal{M}^*(\mathcal{P}^s)$ (resp. $\mathcal{M}^*(\mathcal{P}^t)$). Sometimes we also marginalize over $e_h$ and use $d_h^{s,\pi}(\cdot, \cdot)$ (resp. $d_h^{t,\pi}(\cdot, \cdot)$) to denote the marginal distribution of $s_h, a_h$ on $\mathcal{M}^*(\mathcal{P}^s)$ (resp. $\mathcal{M}^*(\mathcal{P}^t)$). For any value function $g_{h+1} : \mathcal{S} \to \mathbb{R}$ and transition dynamics $P_h(\cdot \mid s_h, a_h, e_h)$, we use $P_h g_{h+1}(s_h, a_h, e_h) \stackrel{\text{def}}{=} \sum_{s'} g_{h+1}(s') P_h(s' \mid s_h, a_h, e_h)$ to denote the expectation of $g_{h+1}$ under $P_h$. For any function space $\mathcal{X}$ with function $x^* \in \mathcal{X}$, define the translated space $\mathcal{X} - x^* \stackrel{\text{def}}{=} \{x - x^* : x \in \mathcal{X}\}$. A comprehensive table of notation, covering both the main body and the appendix, is provided in Appendix A.

## 4. Meta Algorithm and Methodology

We're ready to present our algorithms to learn the optimal policy in the online strategic interaction model. We first introduce a meta-algorithm (i.e., Algorithm 1), then come to its model-based variant tailored for our setting.

**Algorithm 1** Meta Algorithm

1: **Input:** hypothesis class $\mathcal{H}$, confidence level $\beta$, total episodes $K$, auxiliary function class $\mathcal{U}$.
2: Initialize the exploration policy $\pi^1$ as a uniform policy.
3: **for** $k = 1, 2, ..., K$ **do**
4:      Establish dataset $\mathcal{D}_h^k$ with policy $\pi^k$ for $h \in [H]$.
5:      Compute empirical risk function $\mathcal{L}_h^k(c_h)$ for all $c_h \in \mathcal{H}_h$ with auxiliary function class $\mathcal{U}$.
6:      Construct high probability confidence set $\bar{\mathcal{C}}^k = \{c = (c_1, ..., c_H) \in \mathcal{H} : \mathcal{L}_h^k(c_h) \leq \beta, \forall h \in [H]\}$.
7:      Compute $c^{k+1} = \arg\max_{c \in \bar{\mathcal{C}}^k} V_c^*, \pi^{k+1} = \pi_{c^{k+1}}$.
8: **end for**
9: **Output:** $\pi^i$ for $i \sim \text{Unif}([K])$.

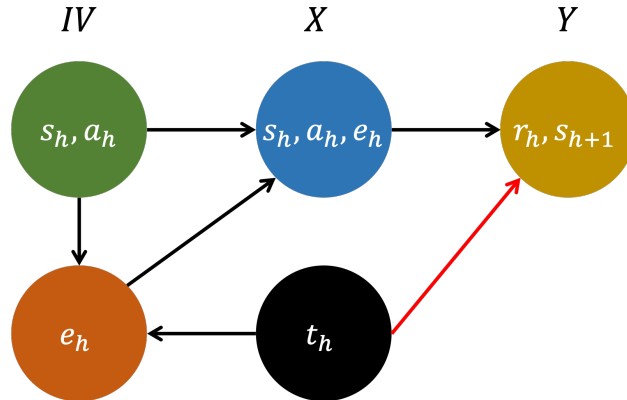

*Figure 2.* The causal graph for the strategic interaction between the principal and the $h$-th agent. The red line denotes the confounding between $(r_h, s_{h+1})$ and $t_h$. "IV" means instrumental variables.

Suppose we have access to a hypothesis class $\mathcal{H} = \{\mathcal{H}_h\}_{h=1}^H$ that realizes the underlying model or equally value functions, we follow the optimism principle in the face of uncertainty to perform efficient exploration by selecting the optimistic hypothesis $c^{k+1}$ and its related policy $\pi_{c^{k+1}}$ (e.g., the optimal policy of $c^{k+1}$ as if $c^{k+1}$ is the model). The high probability confidence set is constructed by the empirical risk function with the help of an auxiliary function class $\mathcal{U}$ (see Line 1 in Algorithm 1) according to the casual identification established through our NPIV method (see Section 4.1).

### 4.1. Detailed Explanation of Methodology

Recall that we face two coupled challenges aiming at learning the optimal policy.

First, we need to **bypass the confounding issues**. Since $\xi_h$ which depends on unobserved type $t_h$ appears in the reward function, the correlation between it and feedback $e_h$ invalidates traditional direct reinforcement learning methods, calling for instrumental variable method. As a simple observation, we know that the principal's state-action pair serves as a valid instrument. Intuitively, $(s_h, a_h)$ affects the agent's reward both directly and indirectly only by affecting the feedback $e_h$ generated from the agent's best response behavior.

Second, we need to **explore unknown environments**. Given the exploration requires accurate estimates of rewards and transitions, these two challenges are coupled and we propose a corresponding optimism principle generalizing the application of optimistic planning. We visualize the causal graph and our underlying intuition on instrumental variables in Figure 2.

We propose a model-based variant of the meta-algorithm, i.e., OPME-G in Algorithm 2, in our setup, where the hypothesis class $\mathcal{H}_h$ is initialized to the model class $\mathcal{H}_h = (\mathcal{R}_h, \mathcal{P}_h)$, where $\mathcal{R}_h = \{R_h : R_h(s_h, a_h, e_h) \in \mathbb{R}\}$ and $\mathcal{P}_h = \{P_h : P_h(\cdot \mid s_h, a_h, e_h) \in \Delta(\mathcal{S})\}$. Please refer to

Appendices D and H for details on hyperparameter selection and additional information on another variant OPME-D.

It is challenging to estimate $R_h^*$ and $P_h^*$ from the model class because of the confounding issue. We take $R_h^*$ as an example to explain this issue. Suppose the principal receives a sampled reward $r_h$ at state $s_h$ after taking action $a_h$ and observing feedback $e_h$, one may hope $\mathbb{E}[r_h \mid s_h, a_h, e_h] = R_h^*(s_h, a_h, e_h)$ so as to estimate $R_h^*$ by standard least square regression. However, this is not the case with the existence of confounders.

Given the endogenous noise $\xi_h$ in $r_h = R_h^*(s_h, a_h, e_h) + \xi_h$, Equation (3.3) implies that

$$\mathbb{E}_{\mathcal{M}^*(\mathcal{P}^s)}[r_h \mid s_h, a_h, e_h] \neq R_h^*(s_h, a_h, e_h).$$

Here $\mathcal{M}^*(\mathcal{P}^s)$ is the model under source distribution (cf. Section 3). Motivated by Yu et al. (2022); Angrist & Imbens (1995); Angrist & Krueger (2001); Chen & Qi (2022), we turn to tailoring the NPIV method and finally resolve this issue.

**Nonparametric instrumental variable model.** Since $\xi_h$ is a zero-mean noise, we can use $(s_h, a_h)$ as an instrumental variable for $(s_h, a_h, e_h)$ and $r_h$ in the sense that (cf. Appendix F)

$$\mathbb{E}_{\mathcal{M}^*(\mathcal{P}^s)}[r_h - R_h^*(s_h, a_h, e_h) \mid s_h, a_h] = 0. \quad (4.1)$$

Given any roll-in policy $\pi$, the conditional moment equation (i.e., Equation (4.1)) gives

$$\mathbb{E}_{s_h, a_h \sim d_h^{s,\pi}}\big[\mathbb{E}_{e_h}[r_h - R_h^*(s_h, a_h, e_h) \mid s_h, a_h]\big] = 0,$$

which inspires us to estimate $R_h^*$ via the least-square loss in

**Algorithm 2** Optimistic Planning with Minimax Estimation-Dynamical/General (OPME-D/G)

1: **Input:** model class $\mathcal{R}, \mathcal{P}$, confidence level $\beta$ (see Equation (H.4)), number of episodes $K$, discriminator class $\mathcal{F}$, discriminator class $\mathcal{G}$ (used only in OPME-G).
2: Initialize the dataset $\mathcal{D}_h = \emptyset$ for all $h \in [H]$.
3: Initialize the exploration policy $\pi^1$ as a uniform policy.
4: **for** $k = 1, 2, ..., K$ **do**
5:    Roll out $\pi^k$ and collect $\tau^k = \{(s_h^k, a_h^k, e_h^k, r_h^k)\}_{h=1}^H$.
6:    **for** $h = 1, 2, ..., H$ **do**
7:       Add the sample $(s_h^k, a_h^k, e_h^k, r_h^k, s_{h+1}^k)$ to $\mathcal{D}_h$.
8:       Define the empirical risk function $\hat{L}_h^k(\cdot)$ of rewards $R_h \in \mathcal{R}_h$ by Equation (4.4).
9:       Compute confidence set $\mathcal{R}_h^k$ by Equation (4.7).
10:      Define the empirical risk function $\hat{L}_h^k(\cdot)$ of transition dynamics $\boldsymbol{G}_h \in \mathcal{P}_h$ by Equation (D.4) for OPME-D, or $P_h \in \mathcal{P}_h$ by Equation (4.6) for OPME-G.
11:      Compute confidence set $\mathcal{P}_h^k$ by Equation (D.5) for OPME-D, or by Equation (4.8) for OPME-G.
12:    **end for**
13:    Construct model class $\bar{\mathcal{C}}^k$ by Equation (D.6) for OPME-D, or by Equation (4.9) for OPME-G.
14:    Set $\bar{\mathcal{M}}^{k+1} = \arg\max_{\bar{\mathcal{M}} \in \bar{\mathcal{C}}^k} V_{\bar{\mathcal{M}}}^\pi(s_1)$ and $\pi^{k+1} = \pi^*_{\bar{\mathcal{M}}^{k+1}}$.
15: **end for**
16: **Output:** $\pi^i$ for $i \sim \text{Unif}([K])$.

episode $k$, that is,

$$\arg\min_{R_h} \sum_{\tau=1}^k \mathbb{E}_{s_h, a_h \sim d_h^{s, \pi^\tau}} [(\mathbb{E}_{e_h} [r_h - R_h(s_h, a_h, e_h) \mid s_h, a_h])^2].$$
$$(4.2)$$

However, this conditional least-square regression cannot be directly estimated due to the conditional expectation inside the square. Instead, we use the minimax estimation with a discriminator function class $\mathcal{F}_h : \mathcal{S} \times \mathcal{A} \to \mathbb{R}$ in place of Equation (4.2) by Fenchel-Rockafellar duality (Dai et al., 2018; Nachum & Dai, 2020) and we define the risk function $L_h^k$ as

$$L_h^k(R_h) \stackrel{\text{def}}{=} \max_{f_h \in \mathcal{F}_h} l_h^k(R_h, f_h) - \frac{1}{2} \sum_{\tau=1}^k \mathbb{E}_{s, a \sim d_h^{s, \pi^\tau}} [f_h^2(s, a)],$$
$$(4.3)$$

where $l_h^k(R, f) \stackrel{\text{def}}{=} \sum_{\tau=1}^k \mathbb{E}_{s_h, a_h, e_h \sim d_h^{s, \pi^\tau}} [f(s_h, a_h)(R(s_h, a_h, e_h) - r_h)]$.

In each episode $k$, we roll out a trajectory $\tau^k = (s_1^k, a_1^k, e_1^k, r_1^k, ..., s_H^k, a_H^k, e_H^k, r_H^k)$ by $\pi^k$ and construct a dataset $\mathcal{D}_h^k = \{\tau^1, \tau^2, ..., \tau^k\}$. Then we can construct an

empirical version of $L_h^k$, say

$$\hat{L}_h^k(R_h) \stackrel{\text{def}}{=} \max_{f_h \in \mathcal{F}_h} \hat{l}_h^k(R_h, f_h) - \frac{1}{2} \sum_{\tau=1}^k f_h^2(s_h^\tau, a_h^\tau), \ \hat{l}_h^k(R, f)$$

$$\stackrel{\text{def}}{=} \sum_{\tau=1}^k f(s_h^\tau, a_h^\tau) (R(s_h^\tau, a_h^\tau, e_h^\tau) - r_h^\tau). \qquad (4.4)$$

Note that although the minimax estimation is motivated by Yu et al. (2022) which has access to i.i.d. samples for estimation, we can only process a non-i.i.d. but sequentially constructed dataset $\mathcal{D}_h^k$. It turns out that we need to resort to a different and more intricate fast martingale concentration analysis with a new construction of confidence set (cf. Equations (4.7) and (4.8)).

The issue similarly affects the estimation of the transition function $P_h^*$. Given $s_h, a_h$ and $e_h$, the confounding between the random variable $s_{h+1}$ and $e_h$ impedes standard model-based estimation methods (Agarwal et al., 2020; Modi et al., 2021; Liu et al., 2022a). Nonetheless, the approach of value target regression (Ayoub et al., 2020; Zhou et al., 2021) inspires us and has been proven beneficial in addressing this confounding issue with meticulously tailored. We utilize an additional discriminator class $\mathcal{G}$, which is designed to encapsulate the optimal value functions of all candidate models. Specifically, for any $g_{h+1}$ within $\mathcal{G}_{h+1}$ and for any observed $s_h, a_h$, it holds that

$$\mathbb{E}_{\mathcal{M}^*(\mathcal{P}^s)} [g_{h+1}(s_{h+1}) - P_h^* g_{h+1}(s_h, a_h, e_h) \mid s_h, a_h] = 0,$$
$$(4.5)$$

thereby aiding in the effective resolution of confounding variables. This innovative approach allows for more accurate estimations of the transition dynamics by leveraging the capabilities of $\mathcal{G}$ to discriminate among the potential influences of confounders. Note that the conditional moment equation has the same form as the previous one, namely, Equation (4.1). Therefore, we can similarly define the empirical risk function of $P_h$ as

$$\hat{L}_h^k(P_h) \stackrel{\text{def}}{=} \max_{g_{h+1} \in \mathcal{G}_{h+1}} \hat{l}_h^k(P_h, g_{h+1}), \qquad (4.6)$$

$$\hat{l}_h^k(P_h, g_{h+1}) \stackrel{\text{def}}{=} \max_{f_h \in \mathcal{F}_h} \hat{l}_h^k(P_h, g_{h+1}, f_h) - \frac{1}{2} \sum_{t=\tau}^k f_h^2(s_h^\tau, a_h^\tau),$$

where $\hat{l}_h^k(P, g, f) \stackrel{\text{def}}{=} \sum_{\tau=1}^k f(s_h^\tau, a_h^\tau)(Pg(s_h^\tau, a_h^\tau, e_h^\tau) - g(s_{h+1}^\tau))$.

**Knowledge transfer and optimistic planning.** We can estimate $R^*, P^*$ with consistent causal identification discussed above. Leveraging the idea of unsupervised domain adaptation (Ganin et al., 2016), we then use such estimators to guide the exploration process given the target population by optimistic planning.

To be more specific, we construct the high probability confidence set in episode $k$ as

$$\mathcal{R}_h^k \stackrel{\text{def}}{=} \left\{ R_h \in \mathcal{R}_h : \hat{L}_h^k(R_h) \leq \beta_1 \right\}, \qquad (4.7)$$

$$\mathcal{P}_h^k \stackrel{\text{def}}{=} \left\{ P_h \in \mathcal{P}_h : \hat{L}_h^k(P_h) \leq \beta_2 \right\}. \qquad (4.8)$$

The confidence level $\beta_1, \beta_2$ are defined in Equation (H.4). For all $h$ and any $R_h \in \mathcal{R}_h^k$ and $P_h \in \mathcal{P}_h^k$, an aggregated model $\bar{\mathcal{M}}(R, P)$ under the target distribution $\mathcal{P}^{\text{t}}$ with $R = \{R_h\}_{h=1}^H, P = \{P_h\}_{h=1}^H$ is defined by Equations (3.1) and (3.2), which enables us to transfer the learned causal knowledge from the source population to the target population. Hence, we define the confidence set on aggregated models as

$$\bar{\mathcal{C}}^k \stackrel{\text{def}}{=} \left\{ \bar{\mathcal{M}}(R, P) : R_h \in \mathcal{R}_h^k, P_h \in \mathcal{P}_h^k, \forall h \in [H] \right\}. \qquad (4.9)$$

Finally, we choose the exploration policy $\pi^{k+1}$ for episode $k + 1$ as the optimal policy of the model with the highest estimated cumulative reward in $\bar{\mathcal{C}}^k$. Simply put, use $\mathcal{P}^{\text{s}}$ for estimation, use $\mathcal{P}^{\text{t}}$ for exploration!

## 5. Theoretical Results and Analysis

In this section, we outline the assumptions made regarding the function classes and detail the sample complexity of our algorithms.

### 5.1. Necessary Assumptions

**Realizability.** First of all, we require all the function classes to be realizable.

**Assumption 5.1** (Realizability). There exists a constant $B$ such that for any $h \in [H]$,

- $R_h^* \in \mathcal{R}_h, P_h^* \in \mathcal{P}_h$.

- The auxiliary function class $\mathcal{F}_h : \mathcal{S} \times \mathcal{A} \rightarrow \mathbb{R}$ with range $[-B, B]$ captures all required functions, say $\mathbb{E}_{t \sim \mathcal{P}_h^{\text{s}}, e \sim F_h(\cdot|s_h, a_h, t)}[\nu_h(s_h, a_h, e)] \in \mathcal{F}_h$ for any $\nu_h \in \{\mathcal{R}_h - R_h^*, (\mathcal{P}_h - P_h^*)\mathcal{G}_{h+1}\}$.[3]

- The auxiliary function class $\mathcal{G}_h : \mathcal{S} \rightarrow [-B, B]$ captures all optimal value functions, say $\bar{V}_{\bar{\mathcal{M}}(R,P),h}^* \in \mathcal{G}_h$ of the aggregated model $\bar{\mathcal{M}}(R, P)$ for any $R \in \mathcal{R}, P \in \mathcal{P}$.

Realizability is necessary since the identification is performed under general nonparametric function classes. For

---

[3] Recall that $(\mathcal{P}_h - P_h^*)\mathcal{G}_{h+1} = \{P_h g_{h+1} - P_h^* g_{h+1} : P_h \in \mathcal{P}_h, g_{h+1} \in \mathcal{G}_{h+1}\}$.

example, the second assumption requires discriminator function class $\mathcal{F}_h$ can capture the projection of $\nu_h$ from the input space to the instrumental variable space in order to bound the error of corresponding minimax estimation.

**Ill-posedness measure.** We overcome the confounding issue through the NPIV model to build a consistent estimator of the model via conditional moment equations (i.e., Equations (4.1) and (4.5)). Taking the reward functions as an example, these equations enable us to bound the projected mean square error (MSE) of $R_h$, namely,

$$\mathbb{E}_{s_h, a_h \sim d_h^{\text{s}, \pi}} \left[ \mathbb{E}_{e_h} \left[ (R_h - R_h^*)(s_h, a_h, e_h) \mid s_h, a_h \right]^2 \right],$$

while the sample complexity is associated with the MSE

$$\mathbb{E}_{s_h, a_h, e_h \sim d_h^{\text{s}, \pi}} \left[ (R_h(s_h, a_h, e_h) - R_h^*(s_h, a_h, e_h))^2 \right].$$

Bridging the gap between projected MSE and MSE leads to an ill-posed inverse problem. People quantify the difficulty of this kind of problems via the following ill-posedness measure.

**Definition 5.2.** Define the ill-posedness measure $\tau_h$ as

$$\tau_h \stackrel{\text{def}}{=} \max_{\nu_h, \pi} \frac{\mathbb{E}_{s_h, a_h, e_h \sim d_h^{\text{s}, \pi}} \left[ \nu_h^2(s_h, a_h, e_h) \right]}{\mathbb{E}_{s_h, a_h \sim d_h^{\text{s}, \pi}} \left[ \mathbb{E}_{e_h} \left[ \nu_h(s_h, a_h, e_h) \mid s_h, a_h \right]^2 \right]},$$

where the maximum of $\nu_h$ is taken over $\nu_h \in \{\mathcal{R}_h - R_h^*, (\mathcal{P}_h - P_h^*)\mathcal{G}_{h+1}\}$.

The ill-posedness measure is widely used in economics to quantify estimation errors of related instrumental variable models (Dikkala et al., 2020; Chen & Qi, 2022; Yu et al., 2022). It illustrates how confounders increase the difficulty of the learning problem and are inevitable in sample complexity.

**Knowledge transfer error.** Apparently, the knowledge transfer is infeasible when the target population drastically differs from the source population where the online data is collected (Pearl & Bareinboim, 2011). We quantify the error caused by transferring knowledge as follows.

**Definition 5.3.** Define the knowledge transfer multiplicative term $C_h^{\text{f}}$ as

$$C_h^{\text{f}} \stackrel{\text{def}}{=} \max_{\nu_h, \pi} \frac{\mathbb{E}_{s_h, a_h, e_h \sim d_h^{\text{t}, \pi}} \left[ \nu_h^2(s_h, a_h, e_h) \right]}{\mathbb{E}_{s_h, a_h, e_h \sim d_h^{\text{s}, \pi}} \left[ \nu_h^2(s_h, a_h, e_h) \right]},$$

where $\nu_h \in \{\mathcal{R}_h - R_h^*, (\mathcal{P}_h - P_h^*)\mathcal{G}_{h+1}\}$.

The multiplicative term $C_h^{\text{f}}$ quantifies the distributional shift when transferring estimators computed from online data to the target population. In the literature, it is often referred

to as "Concentrability" (Foster et al., 2021b). Concentrability is a simple but fairly strong notion of coverage, which requires that the distribution of our data collection evenly covers the target distribution we are interested in. It captures the effort needed to bypass the knowledge transfer.

## 5.2. Sample Complexity Results

We now introduce the sample complexity of our algorithms.

**Theorem 5.4.** *Under Assumption 5.1, with probability at least $1 - \delta$, the sample complexity of the model-based algorithm to learn an $\epsilon$-optimal policy is bounded by*

$$\tilde{O}\left(\sum_{h=1}^{H} B^2 d_{\mathrm{V},h} \tau_h C_h^{\mathrm{f}} \log(|\mathcal{R} \times \mathcal{P} \times \mathcal{G} \times \mathcal{F}|/\delta) \cdot \epsilon^{-2}\right).$$

*Here $d_{\mathrm{V},h}$ is the distributional Eluder dimension of the model class (cf. Appendix E).*

We use a simple example to estimate the magnitude of the Eluder dimension and defer further discussion to Appendix E.

*Example* 5.5 (Linear MDPs (Jin et al., 2021)). Consider the $d$-dimensional linear function classes

$$\mathcal{R}_h = \left\{\theta_h \in \mathbb{R}^d : R_h(s_h, a_h, e_h) = \phi^\top(s_h, a_h, e_h)\theta_h\right\},$$

$$\mathcal{P}_h = \left\{\psi_h \in \mathbb{R}^d : P_h(s_h, a_h, e_h) = \phi^\top(s_h, a_h, e_h)\psi_h\right\},$$

where $\phi$ is a known $d$-dimensional feature mapping. We assume that the underlying reward function is $R_h^*(s_h, a_h, e_h) = \phi^\top(s_h, a_h, e_h)\theta_h^*$ for an unknown $\theta_h^*$ while the true transition kernel is $P_h^*(s_h, a_h, e_h) = \phi^\top(s_h, a_h, e_h)\psi_h^*$ for some unknown $\psi_h^*$. Then, we know that $d_{\mathrm{V},h} \lesssim \tilde{O}(d)$, yielding $\tilde{O}\left(\sum_{h=1}^{H} B^2 d\tau_h C_h^{\mathrm{f}} \log(|\mathcal{R} \times \mathcal{P} \times \mathcal{G} \times \mathcal{F}|/\delta) \cdot \epsilon^{-2}\right)$ sample complexity to learn a $\epsilon$-optimal policy.

Theorem 5.4 shows the sample complexity to learn an $\epsilon$-optimal policy in such a general MDP setting scales as $\tilde{O}(1/\epsilon^2)$ with a linear dependency on the other terms, including $d_{\mathrm{V},h}$, $\tau_h$ and $C_h^{\mathrm{f}}$. The necessity of the ill-posed measure and knowledge transfer multiplicative term is discussed in Appendix I.

Lastly, we end this section with a remark on relevant lower bounds.

*Remark* 5.6. Domingues et al. (2021) proves that an $\tilde{O}(\epsilon^{-2} \cdot \log(1/\delta))$ sample complexity is inevitable to identify an $\epsilon$-optimal policy with probability $1 - \delta$ even without confounding issues. Hence, we conclude that our algorithms achieve optimal, though omitting logarithmic terms, sample complexity concerning $\epsilon$. The dependence of lower bounds on problem-dependent parameters is of independent interest, and we leave it as a future research avenue.

## 6. Discussion and Conclusion

In this paper, we explore the theoretical impacts of information asymmetry and knowledge transferability within the context of reinforcement learning. Specifically, we conceptualize these interactions using an online strategic interaction model, framed through a generalized principal-agent problem, and we introduce algorithms designed to determine the principal's optimal policy within a MDP framework. We introduce innovative estimation techniques utilizing the NPIV method to address confounding issues stemming from information asymmetry. Additionally, we quantify the errors associated with knowledge transfer, providing a robust framework for understanding these dynamics in complex environments. Finally, we propose an algorithm with $\tilde{O}(1/\epsilon^2)$ sample complexity and show its optimality with respect to the optimality gap $\epsilon$.

Nevertheless, our model also has some limitations. For example, it requires the target distribution $\mathcal{P}^t$ and the feedback manipulation distribution $F$ to be known to the principal. There are two considerations on why we cannot remove this condition:

- The optimal policy directly depends on them but we assume no online interaction ability on the target population. That is to say, we need at least some (passive) samples on the target population to estimate the target distribution (e.g., by clustering). Also, it's a standard assumption in Myerson's auction and coordination theory of principal-agent problems (Myerson, 1981; 1982; Gan et al., 2022a) that the target type distribution is known.

- The feedback manipulation distribution is often relatively easy to determine in reality. In many cases, it is either a canonical distribution known to the public or a predetermined quantity according to the principal marketing research on the target population.

Questions raise themselves for future explorations. We find that the states of such decision-making problems are not always fully observable (Brown & Sandholm, 2018; 2019; Futoma et al., 2020). For example, part of the attributes of a third domain such as the government or market are unknown to both the principal and the agents. Is it possible to solve reinforcement learning problems with confounding issues under partially observable Markov decision processes (POMDPs)? Is the NPIV method still valid with value bridge functions (Shi et al., 2021; Uehara et al., 2022a;b)? We leave these interesting questions as potential next steps.

## Impact Statement

This paper presents work whose goal is to advance the field of Machine Learning. There are many potential societal

consequences of our work, none which we feel must be specifically highlighted here.

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

# A. Notation Table

For the convenience of the reader, we summarize the notations in the paper as a notation table.

| Notation | Explanation |
|---|---|
| $\mathcal{S}, \mathcal{A}, P, R, H$ | parameters of an online strategic interaction model |
| $s_h, a_h, e_h, t_h$ | state, action, feedback, private type at step $h$ |
| $\mathcal{Z}_h, \bar{\mathcal{Z}}_h$ | the space of $M$-step history ($\bar{\mathcal{Z}}_h$ excludes the state $s_h$) |
| $z_h, \bar{z}_h$ | an $M$-step history in $\mathcal{Z}_h, \bar{\mathcal{Z}}_h$ |
| $\mathcal{P}^{\mathrm{s}} = (\mathcal{P}_1^{\mathrm{s}}, \mathcal{P}_2^{\mathrm{s}}, ..., \mathcal{P}_H^{\mathrm{s}})$ | the distribution of private types on source population |
| $\mathcal{P}^{\mathrm{t}} = (\mathcal{P}_1^{\mathrm{t}}, \mathcal{P}_2^{\mathrm{t}}, ..., \mathcal{P}_H^{\mathrm{t}})$ | the distribution of private types on target population |
| $R_h^{\mathrm{a}}(s_h, a_h, t, b)$ | private reward function of the agent with type $t$ and action $b$ |
| $b_h = \arg\max_b R_h^{\mathrm{a}}(s_h, a_h, t_h, b)$ | private action of the $h$-th agent |
| $\tilde{F}_h(\cdot \mid s_h, a_h, t_h, b_h)$ | feedback manipulation distribution |
| $F_h(\cdot \mid s_h, a_h, t_h)$ | $\tilde{F}_h(\cdot \mid s_h, a_h, t_h, \arg\max_b R_h^{\mathrm{a}}(s_h, a_h, t_h, b))$, equivalent to $\tilde{F}_h$ |
| $\mathcal{M}^*(\mathcal{P}^{\mathrm{s}}), \mathcal{M}^*(\mathcal{P}^{\mathrm{t}})$ | the online strategic interaction model under the source/target population |
| $d_h^{\mathrm{s},\pi}(s_h, a_h, e_h), d_h^{\mathrm{t},\pi}(s_h, a_h, e_h)$ | occupancy measure of $s_h, a_h, e_h$ on $\mathcal{M}^*(\mathcal{P}^{\mathrm{s}}), \mathcal{M}^*(\mathcal{P}^{\mathrm{t}})$ |
| $d_h^{\mathrm{s},\pi}(s_h, a_h), d_h^{\mathrm{t},\pi}(s_h, a_h)$ | occupancy measure of $s_h, a_h$ on $\mathcal{M}^*(\mathcal{P}^{\mathrm{s}}), \mathcal{M}^*(\mathcal{P}^{\mathrm{t}})$ |
| $\beta, \beta_1, \beta_2, \beta_3$ | confidence level |
| $\mathcal{R} = (\mathcal{R}_1, ..., \mathcal{R}_H), \mathcal{P} = (\mathcal{P}_1, ..., \mathcal{P}_H)$ | hypothesis model class |
| $R_h^*(s_h, a_h, e_h), P_h^*(s_h, a_h, e_h)$ | underlying reward function and transition function |
| $\bar{P}^*, \bar{R}^*$ | aggregated reward and transition on target population |
| $\bar{\mathcal{M}}^* = (\mathcal{S}, \mathcal{A}, \bar{P}^*, \bar{R}^*, H, s_1)$ | aggregated MDP model on target population |
| $\bar{\pi}^*$ | optimal Markovian policy of $\bar{\mathcal{M}}^*$ |
| $\bar{V}_{\bar{\mathcal{M}}^*}^{\pi}$ | value function of policy $\pi$ on $\bar{\mathcal{M}}^*$ |
| $\xi_h$ | additive endogenous noise of rewards |
| $\mathcal{F} = (\mathcal{F}_1, ..., \mathcal{F}_H), \mathcal{G} = (\mathcal{G}_1, ..., \mathcal{G}_H)$ | discriminator function class |
| $P_h g_{h+1}(s_h, a_h, e_h)$ | Bellman backup $\sum_{s'} g_{h+1}(s') P_h(s' \mid s_h, a_h, e_h)$ |
| $L_h^k, l_h^k$ | population risk function of candidate models |
| $\hat{L}_h^k, \hat{l}_h^k$ | empirical risk function of candidate models |
| $\mathcal{R}_h^k, \mathcal{P}_h^k, \bar{\mathcal{C}}^k$ | confidence set |
| $\tau_h$ | ill-posed measure |
| $C_h^{\mathrm{f}}$ | knowledge transfer multiplicative term |
| $d_{\mathrm{M},h}, d_{\mathrm{V},h}$ | distributional Eluder dimension of the model class |
| $\nu_h$ (main text) | any function in the union space of $\mathcal{R}_h - R_h^*$ and $(\mathcal{P}_h - P_h^*)\mathcal{G}_{h+1}$ |
| $\boldsymbol{G}^* = (\boldsymbol{G}_1^*, ..., \boldsymbol{G}_H^*)$ | dynamical system transition function (cf. Appendix D.1) |
| $d_{\mathrm{s}}$ | the dimension of the state in dynamical system |
| $\mathcal{P}_{h,i}, G_{h,i}(G_{h,i} \in \mathcal{P}_{h,i})$ | dynamical transition class and one function in the class for dim $i \in [d_{\mathrm{s}}]$ |
| $\mathcal{Y}_h$ | the union space of $\mathcal{R}_h - R_h^*, (\mathcal{P}_h - P_h^*)\mathcal{G}_{h+1}, \mathcal{P}_{h,i} - G_{h,i}^*, \forall i \in [d_{\mathrm{s}}]$ |
| $\nu_h$ (appendix) | any function in $\mathcal{Y}_h$ |

# B. Detailed Discussion of Related Work

We discuss some related works in detail in this section.

## B.1. Further Topics and Related Work

In this section, we provide a more in-depth and comprehensive discussion of the related literature.

**Online exploration in MDPs.** Online exploration problem is one of the most fundamental problems in reinforcement learning (Sutton & Barto, 2018). It has been extensively studied in tabular MDPs (Auer et al., 2008; Azar et al., 2017; Dann et al., 2017; Jin et al., 2018; Dann et al., 2019; Zanette & Brunskill, 2019; Zhang et al., 2022), MDPs with linear function approximation (Yang & Wang, 2019; Jin et al., 2020; Yang & Wang, 2020; Cai et al., 2020; Ayoub et al., 2020; Agarwal et al., 2020; Zhou et al., 2021; Hu et al., 2021; Chen et al., 2021) and general function approximation (Russo & Van Roy, 2013; Jiang et al., 2017; Sun et al., 2019; Wang et al., 2020; Jin et al., 2021; Du et al., 2021; Foster et al.,

2021a; Zhong et al., 2022; Liu et al., 2022b; Chen et al., 2022). In the standard exploration problem, the agent is required to balance the exploration–actively exploring the unknown part of the environment to discover potential high rewards, and the exploitation–exploiting the discovered area to earn higher rewards. The exploration-exploitation trade-off is known as one of the core challenges in RL with large state and action space. This paper studies the online strategic interactions in RL with general function approximation, which naturally inherits the exploration-exploitation trade-off challenge. Furthermore, the strategic interaction model presents a more challenging exploration issue in that the agent is able to manipulate the distribution of the strategic feedback based on her private type, which is not controlled by the principal. We tackle this issue by leveraging the NPIV model (Ai & Chen, 2003; Newey & Powell, 2003) and optimism in the face of uncertainty principle (Abbasi-Yadkori et al., 2011).

**RL with confounders.** Our work is also related to RL with confounders, as the private type $t_h$ acts as an unobserved confounder in the strategic interaction model. The off-policy evaluation (OPE) problem with unmeasured confounders has been actively studied in recent years (Kallus & Zhou, 2020; Tennenholtz et al., 2020; Shi et al., 2021; Bennett & Kallus, 2021; Bennett et al., 2021; Nair & Jiang, 2021) by leveraging the techniques in casual inference (Pearl, 2009). There are also a number of works studying the policy optimization in offline RL with a confounded dataset (Wang et al., 2021; Liao et al., 2021; Kallus & Zhou, 2021; Lu et al., 2022; Yu et al., 2022; Wang et al., 2022). On the contrary, our work studies the online setting, where the unobserved private type $t_h$ confounds with the rewards and transition dynamics. Among these works, the most related one to us is Yu et al. (2022), which studied the strategic MDPs in the offline setting. Our work is different from theirs in several aspects. First, we study the strategic interactions in the online setting which violates the widely used i.i.d. assumption and calls for new technical solutions. Second, we explore a setup that more accurately simulates the real world, albeit with an added challenge—knowledge transportability. Lastly, we extend their transition class from nonlinear dynamical systems to general transition classes, which can handle noises that are not additive. We refer the readers to Appendix B.2 for a detailed comparison.

**Transfer learning for reinforcement learning domains.** Taylor & Stone (2009) gives an inspiring survey on how to do transfer learning when facing RL scenarios while Zhu et al. (2023) extends it to deep reinforcement learning. As for the application literature, Hua et al. (2021) utilizes transfer learning and reinforcement learning in the field of robotics and Gamrian & Goldberg (2019) uses this idea to solve visual tasks. Yang et al. (2021) considers how to use transfer learning when facing multiagent reinforcement learning. Here, we give some theoretical perspectives on how knowledge transportability affects the sample complexity of reinforcement learning.

## B.2. Novel Analysis Beyond the Offline Strategic MDP and Standard Exploration in RL

We discuss the novelties of our algorithms beyond the PLAN algorithm proposed by Yu et al. (2022) that learns the optimal policy of strategic MDPs under general function approximation in the offline setting, and the exploration algorithms in standard RL literature with general function approximation. The novelties mainly come from the problem settings and theoretical analysis, besides the difference between online exploration and offline planning.

- In terms of the problem settings, we study a generalized version of the original strategic MDPs proposed in Yu et al. (2022), which assumed the transition dynamics belong to a nonlinear dynamical system class with additive Gaussian noises. We show how to learn the optimal policy in general transition classes (cf. Section 4.1) given access to an extra discriminator function class. Moreover, we study a generalized online strategic interaction model that breaks static policy assumptions and needs to balance exploration and nuisance caused by time-varying data. Our analysis, focusing on the utilization of correlated data as instrumental variables, is entirely independent from their work, which introduces a novel design and analysis for a new NPIV model.

- In terms of the theoretical analysis, ours are different from the analysis in Yu et al. (2022) in three aspects: the casual identification, the construction of the confidence set, and the analysis therein.

  Although both our work and their work take advantage of the NPIV model to estimate the empirical risk function $\hat{L}$ (see, e.g., Equation (4.4)), Yu et al. (2022) established the concentration bound with an i.i.d. distributed dataset, but our work employs martingale concentration analysis with Freedman's inequality. Moreover, their work constructed the confidence set by restricting the value difference between a candidate model and the minimizer of $\hat{L}$, while we construct the confidence set by solely restricting the value of $\hat{L}$ of the candidate model (see, e.g., Equation (4.7)). Our analysis shows that our construction is not only cleaner but enables us to **get rid of** the symmetric and star-shaped

assumption imposed on $\mathcal{F}$ (see Assumption 5.3 of Yu et al. (2022)). Moreover, we design cleaner proofs only using the realizability assumption and the boundedness of zeros of a concave quadratic function, simplifying the complicated proof based on the symmetric and star-shaped assumption of $\mathcal{F}$ in Yu et al. (2022).

- In an online strategic interaction model, the principal is asked to interact with a source population of agents for $H$ steps, where each agent has a private type and commits a private action, both unknown to the principal. The goal of the principal is to learn a return-maximizing policy for another population of agents called the target population. The property of the source population is described through the private type distribution of the agents $\mathcal{P}^s$ (the source distribution), and the feedback manipulation distribution $F$. The source distribution is unknown to the principal. For the feedback manipulation distribution $F$, we made a simplification in the paper to assume it is the same for the source population and target population, but we note that this distribution can be different for the source and the target. The principal does not need to know the corresponding $F$ of the source population. This is part of the reason why we cannot reduce the problem to a naive single-agent RL problem since the source distribution and source feedback manipulation distribution are both unknown. Another challenge beyond single-agent RL is the distributional shift between the source distribution and the target distribution. Our model-based approach follows a similar idea of unsupervised domain adaptation where we estimate the underlying model $R^*, P^*$ from the source distribution at first, then transfer the estimator to the target. A key difference to the unsupervised domain adaptation here is the use of target distribution. Standard unsupervised domain adaptation uses target distribution to build importance-sampling estimators. Nevertheless, as we don't know $\mathcal{P}^s$, it's impossible to use importance sampling to construct an unbiased or even consistent estimator. Besides, confounding issues beyond traditional RL problems hamper our use of importance sampling as well. Consequently, we use the target distribution to design the exploration policy for the next episode.

## C. Detailed Descriptions of the Online Strategic Interaction Model and the Motivation

This section offers more details of the online strategic interaction model and the motivations to learn this model.

### C.1. More Details of the Online Strategic Interaction Model

For a strategic interaction model $\mathcal{M}^*(\mathcal{P})$ with reward function $R^*$, transition dynamics $P^*$, and agent private type distribution $\mathcal{P}$, the casual graph of the strategic interactions between the principal and the $h$-th agent is shown in Figure 2. To better illustrate this strategic interaction process, we take the reward $r_h$ as an example. The causal structure of the transition $s_{h+1}$ is the same as $r_h$.

The reward equals

$$r_h = R_h^*(s_h, a_h, e_h) + \xi_h$$

by definition, where $\xi_h$ is an endogenous noise that may be confounded with $t_h$ (Yu et al., 2022; Harris et al., 2022).

**Definition C.1** (Informal, see Pearl (2009) for a formal definition). A factor in a causal model or causal system whose value is determined by the states of other variables in the system is called an endogenous variable. On the contrary, an exogenous variable is a factor in a causal model or causal system whose value is independent of the states of other variables in the system.

According to the definition and observe that $\xi_h$ is only confounded with the private type $t_h$, it holds that

$$\mathbb{E}_{t_h \sim \mathcal{P}} [\xi_h \mid s_h, a_h] = 0. \tag{C.1}$$

Now, let's talk about why Equation (C.1) makes sense. Considering the source distribution $\mathcal{P}^s$, if $\mathbb{E}_{t_h \sim \mathcal{P}_h^s} [\xi_h \mid s_h, a_h] \neq 0$, we can set $R_h^*(s_h, a_h, e_h)$ to be $R_h^*(s_h, a_h, e_h) + \mathbb{E}_{t_h \sim \mathcal{P}_h^s} [\xi_h \mid s_h, a_h]$. Then, this process demeans $\xi_h$ and proves the legitimacy of Equation (C.1), namely, we can assume $\xi_h$ is zero-mean under $\mathcal{P}^s$ without loss of generality. Nevertheless, we in general cannot guarantee that $\mathbb{E}_{t_h \sim \mathcal{P}_h^t} [\xi_h \mid s_h, a_h] = 0$ whenever the target distribution $\mathcal{P}^t$ is deviated from the source distribution $\mathcal{P}^s$. Note that $\xi_h$ doesn't depend on $a_h$ actually, so the optimal policy under dynamics $R_h^*(s_h, a_h, e_h)$ and $R_h^*(s_h, a_h, e_h) + \mathbb{E}_{t_h \sim \mathcal{P}_h^t} [\xi_h \mid s_h, a_h]$ are the same. Consequently, albeit adding the counterfactual assumption that $\mathbb{E}_{t_h \sim \mathcal{P}_h^t} [\xi_h \mid s_h, a_h] = 0$, we will find an identical policy as the case without it regarding the way we form $\bar{\mathcal{M}}^*$. Therefore, we conclude that assuming $\mathbb{E}_{t_h \sim \mathcal{P}_h^t} [\xi_h \mid s_h, a_h] = 0$ won't simplify the problem and will yield equivalent sample complexity.

Thereupon, we keep Equation (C.1) for a more concise presentation and want to remind readers that this assumption is neither necessary nor does it affect the results.

However, this confounding issue causes a severe challenge that when conditional on any $(s_h, a_h, e_h)$

$$\mathbb{E}_{\mathcal{M}^*(\mathcal{P})}\left[r_h \mid s_h, a_h, e_h\right] \neq R_h^*(s_h, a_h, e_h)$$

because the feedback $e_h$ also depends with $t_h$ and the noise term $\xi_h$ will not be zero-mean given $e_h$.

Observe that conditioned on any $(s_h, a_h)$ it holds that

$$
\begin{aligned}
\mathbb{E}_{\mathcal{M}^*(\mathcal{P})}\left[r_h \mid s_h, a_h\right] &= \mathbb{E}_{t_h \sim \mathcal{P}_h, e_h \sim F_h(\cdot \mid s_h, a_h, t_h)}\left[r_h \mid s_h, a_h\right] \\
&= \mathbb{E}_{t_h \sim \mathcal{P}_h, e_h \sim F_h(\cdot \mid s_h, a_h, t_h)}\left[R^*(s_h, a_h, e_h)\right].
\end{aligned}
\tag{C.2}
$$

The second equation holds because of Equation (C.1). The last term in Equation (C.2) is the definition of the aggregate model (i.e., Equations (3.1) and (3.2)).

The transition dynamics have the same causal structure of the rewards, but we cannot express this structure with additive endogenous noise as the rewards. The causal identification of the next state $s_{h+1}$ should be

$$s_{h+1} \sim \mathbb{E}_{t_h \sim \mathcal{P}_h, e_h \sim F_h(\cdot \mid s_h, a_h, t_h)}\left[P^*(s_h, a_h, e_h)\right]$$

according to previous analysis. Therefore, the expected reward and transition are identical for the corresponding strategic interaction model $\mathcal{M}^*(\mathcal{P})$ and the aggregated model.

### C.2. Motivating Examples

We offer two motivating examples here for the MDP setting.

- The non-compliant agents in the recommendation system (Yu et al., 2022). The principal is a company offering some recommendation services, and the agents are clients. The private types $t_h$ are the profiles of the clients (e.g., the category of items they like, and their shopping preferences). The state $s_h$ is a public condition of the company (e.g., the inventory level, the transportation service condition, etc.), and the action $a_h$ is the recommended item to the client. The private action $b_h$ is the item ordered by the client (not necessarily the same as $a_h$), and the feedback $e_h$ is exactly the action $b_h$ in this case. The reward $r_h$ is the real profit made from this client, which is related to the state $s_h$, the feedback $e_h = b_h$, the transportation fee, the willingness to buy accessories, etc. In this case, the reward distribution is highly correlated to the private type $t_h$, and such correlation can not be expressed by a simple model $R(s_h, a_h, e_h)$ with an independent additive noise. That explains the necessity to introduce a confounding noise model and thus endogenous variables. Here, $H = 12$ represents 12 months in one year periodically.

- The admission of university (Harris et al., 2022). A university (the principal) is making admission decisions for a population of applicants (the agents). Typically, there are several, namely $H$, rolling admissions decisions throughout the year. The action $a_h$ of the university is an assessment rule that measures some predicted outcome (such as the overall GPA) of the students if admitted. The predicted outcome is also used as the reward $r_h$ for the university. The feedback $e_h$ should be some observed/measured attributes of the students, like the standardized test scores, high school GPA, etc. The students try to maximize the alignment between the observed attributes $e_h$ and the assessment rule $a_h$ through their efforts $b_h$, which is unobservable. The private type $t_h$ of the student can be many (unobserved) attributes related to $e_h$ or $r_h$ (the predicted outcome of the student), such as the baseline scores the students can get without any efforts, the efficiency of changing efforts to improved scores, etc. The noise term $\xi_h$ summarizes all other environmental factors that can impact the agent's true outcome when we control for observable attributes, such as the effect of the institutional barriers the student faces on their actual college GPA. The noise term $\xi_h$ can be arbitrarily correlated with private type $t_h$, which also impacts the feedback $e_h$.

## D. Complete Algorithm in the Online Strategic Model

The complete algorithm, Optimistic Planning with Minimax Estimation (OPME), is presented as Algorithm 2. We provide OPME for two types of transition classes: the general transition class (cf. Section 4.1) and the dynamical transition class (see Appendix D.1). The algorithm OPME-General (OPME-G) is the complete version of the model-based algorithm described in Section 4.1.

## D.1. The Dynamical Transition Class

Following the setting of Yu et al. (2022), we also study a restricted transition class that $P_h$ is a non-linear dynamical system. Suppose $\mathcal{S} \subseteq \mathbb{R}^{d_{\rm s}}$ and

$$s_{h+1} = \boldsymbol{G}_h^*(s_h, a_h, e_h) + \eta_h$$

for some unknown $\boldsymbol{G}_h^*$, where $\eta_h$ is a $d_{\rm s}$-dimensional zero-mean noise possibly confounded with $t_h$. Thus, we assume the noise term $\eta_h$ can be decomposed to a correlation term and an independent noise

$$\eta_h = \text{corr}(t_h) + \eta_h', \tag{D.1}$$

where $\text{corr}(\cdot)$ is a correlation function and $\eta_h'$ is an independent standard Gaussian noise $\eta_h' \sim \mathcal{N}(\boldsymbol{0}, \boldsymbol{I}_{d_{\rm s}})$.

In such transition classes, $\mathcal{P}_h$ can be written as $\mathcal{P}_h = \{\boldsymbol{G}_h : \boldsymbol{G}_h = (G_{h,1}, G_{h,2}, ..., G_{h,d_{\rm s}}) \in \mathbb{R}^{d_{\rm s}}\}$. For simplicity, we assume $\mathcal{P}_h$ can be decomposed as the product of $d_{\rm s}$ classes $\mathcal{P}_h = \mathcal{P}_{h,1} \times \mathcal{P}_{h,2} \times \cdots \times \mathcal{P}_{h,d_{\rm s}}$, where

$$\mathcal{P}_{h,i} = \{G_{h,i} : G_{h,i}(s_h, a_h, e_h) \in \mathbb{R}\} \tag{D.2}$$

denotes the transition class for the $i$-th coordinate and step $h$. Note that $G_{h,i}^* \in \mathcal{P}_{h,i}$ for each $i \in [d_{\rm s}]$.

We propose the algorithm OPME-Dynamical (OPME-D, Algorithm 2) that learns the optimal policy of $\mathcal{M}^*(\mathcal{P}^{\rm t})$ under the dynamical system transition class.

Similar to the reward function, the identification of the transition function can also be established by the NPIV method, that is

$$\mathbb{E}_{\mathcal{M}^*(\mathcal{P}^{\rm s})}\left[s_{h+1} - \boldsymbol{G}_h^*(s_h, a_h, e_h) \mid s_h, a_h\right] = \boldsymbol{0}. \tag{D.3}$$

For $i \in [d_{\rm s}]$, we can construct the empirical risk function $\hat{L}_h^k(G_{h,i})$ for transition $G_{h,i}$ analogously. With a little abuse of notations, we define

$$\hat{L}_h^k(G_{h,i}) \overset{\text{def}}{=} \max_{f_h \in \mathcal{F}_h} \hat{l}_h^k(G_{h,i}, f_h) - \frac{1}{2}\sum_{\tau=1}^{k} f_h^2(s_h^\tau, a_h^\tau), \tag{D.4}$$

and

$$\hat{l}_h^k(G_{h,i}, f_h) \overset{\text{def}}{=} \sum_{\tau=1}^{k} f_h(s_h^\tau, a_h^\tau)\left(G_{h,i}(s_h^\tau, a_h^\tau, e_h^\tau) - s_{h+1,i}^\tau\right).$$

The high probability confidence set in episode $k$ for the transition function is defined as

$$\mathcal{P}_{h,i}^k \overset{\text{def}}{=} \left\{G_{h,i} \in \mathcal{P}_{h,i} : \hat{L}_h^k(G_{h,i}) \leq \beta_3\right\}, \forall i \in [d_{\rm s}]. \tag{D.5}$$

Here $\beta_3$ (defined in Equation (H.4)) is the confidence level used in OPME-D. For all $h$ and any $R_h \in \mathcal{R}_h^k$ and $\boldsymbol{G}_h \in \mathcal{P}_h^k \overset{\text{def}}{=} \prod_{i=1}^{d_{\rm s}} \mathcal{P}_{h,i}$,[4] we can construct the aggregated model $\bar{\mathcal{M}}(R, \boldsymbol{G})$ under the target distribution $\mathcal{P}^{\rm t}$ with $R = \{R_h\}_{h=1}^H, \boldsymbol{G} = \{\boldsymbol{G}_h\}_{h=1}^H$ by Equations (3.1) and (3.2). Hence, we define the confidence set on aggregated models as

$$\bar{\mathcal{C}}^k \overset{\text{def}}{=} \left\{\bar{\mathcal{M}}(R, \boldsymbol{G}) : R_h \in \mathcal{R}_h^k, \boldsymbol{G}_h \in \mathcal{P}_h^k, \forall h \in [H]\right\}. \tag{D.6}$$

We choose the exploration policy for episode $k+1$ as the optimistic policy with the highest value in $\bar{\mathcal{C}}^k$.

# E. The Distributional Eluder Dimension

**Definition E.1** ($\epsilon$-independence between distributions, Definition 6 of Jin et al. (2021)). Let $\mathcal{G}$ be a function class defined on $\mathcal{X}$, and $\nu, \mu_1, \mu_2, ..., \mu_n$ be probability distributions over $\mathcal{X}$. We call $\nu$ is $\epsilon$-independent of $\mu_1, \mu_2, ..., \mu_n$ with respect to $\mathcal{G}$ if there exists $g \in \mathcal{G}$ such that $\sqrt{\sum_{i=1}^{n}(\mathbb{E}_{\mu_i}[g])^2} \leq \epsilon$, but $|\mathbb{E}_\nu[g]| > \epsilon$.

---

[4] We use $\prod_{i=1}^{d_{\rm s}} \mathcal{P}_{h,i}^k$ as the shorthand for $\mathcal{P}_{h,1}^k \times \mathcal{P}_{h,2}^k \times \cdots \times \mathcal{P}_{h,d_{\rm s}}^k$.

**Definition E.2** (Distributional Eluder Dimension, Definition 7 of Jin et al. (2021)). Let $\mathcal{G}$ be a function class defined on $\mathcal{X}$, and $\Pi$ be a class of distributions over $\mathcal{X}$. The distributional Eluder dimension $\dim_{\mathrm{DE}}(\mathcal{G}, \Pi, \epsilon)$ is defined as the length of the longest sequence $\mu_1, \mu_2, ..., \mu_n$ such that $\mu_i \in \Pi, \forall i \in [n]$ and there exists $\epsilon' \geq \epsilon$ where $\mu_i$ is $\epsilon'$-independent of $\mu_1, ..., \mu_{i-1}$ for all $i \in [n]$.

We define the distribution class $\Pi = \{\Pi_h\}_{h=1}^H$ where $\Pi_h$ collects all the density measures $d_h^{\mathrm{t}, \pi}$ for the optimal policy $\pi$ of any aggregated model $\bar{\mathcal{M}}(R, P), R \in \mathcal{R}, P \in \mathcal{P}$.

For the dynamical system transition class, with a little abuse of notations, we define the distributional Eluder dimension $d_{\mathrm{M}}$ of the model class $(\mathcal{R}, \mathcal{P})$ by

$$d_{\mathrm{M}, h} \overset{\text{def}}{=} \dim_{\mathrm{DE}}(\mathcal{R}_h - R_h^*, \Pi_h, 1/\sqrt{K}) + \sum_{i=1}^{d_{\mathrm{s}}} \dim_{\mathrm{DE}}(\mathcal{P}_{h,i} - P_{h,i}^*, \Pi_h, 1/\sqrt{K}).$$

For the general transition class, we define the distributional Eluder dimension $d_{\mathrm{V}}$ of the model class $(\mathcal{R}, \mathcal{P})$ with respect to the discriminator function class $\mathcal{G}$ as

$$d_{\mathrm{V}, h} \overset{\text{def}}{=} \dim_{\mathrm{DE}}(\mathcal{R}_h - R_h^*, \Pi_h, 1/\sqrt{K}) + \dim_{\mathrm{DE}}(\mathcal{P}_h \mathcal{G}_{h+1} - P_h^* \mathcal{G}_{h+1}, \Pi_h, 1/\sqrt{K}).$$

The distributional Eluder dimensions are small for several model classes (Jin et al., 2021; Jiang et al., 2017). Here we provide a detailed example.

Consider the $d$-dimensional linear function class

$$\mathcal{R}_h = \left\{ \theta_h \in \mathbb{R}^d : R_h(s_h, a_h, e_h) = \phi^\top(s_h, a_h, e_h)\theta_h \right\},$$

where $\phi$ is a known $d$-dimensional feature mapping. The underlying reward function is $R_h^*(s_h, a_h, e_h) = \phi^\top(s_h, a_h, e_h)\theta_h^*$ for an unknown $\theta_h^*$. Then for any $R_h \in \mathcal{R}_h$, there exists a feature function $\psi_h : \Pi \to \mathbb{R}^d$ such that

$$\mathbb{E}_{s_h, a_h, e_h \sim d_h^{\mathrm{t}, \pi}} \left[ R_h(s_h, a_h, e_h) - R_h^*(s_h, a_h, e_h) \right] = \langle \psi_h(\pi), \theta_h - \theta_h^* \rangle, \tag{E.1}$$

where

$$\psi_h(\pi) = \mathbb{E}_{s_h, a_h, e_h \sim d_h^{\mathrm{t}, \pi}} \left[ \phi(s_h, a_h, e_h) \right].$$

According Section C.1 of Jin et al. (2021) and Equation (E.1), we know

$$\dim_{\mathrm{DE}}(\mathcal{R}_h - R_h^*, \Pi_h, 1/\sqrt{K}) \leq \tilde{O}(d)$$

as long as $\|\phi\|_2, \|\theta_h\|_2$ have a uniform upper bound.

Similarly, we can bound the distributional Eluder dimension of $\mathcal{P}_{h,i}$ if

$$\mathcal{P}_{h,i} = \left\{ \bar{\theta}_h \in \mathbb{R}^d : P_{h,i}(s_h, a_h, e_h) = \phi^\top(s_h, a_h, e_h)\bar{\theta}_h \right\}.$$

The distributional Eluder dimension is bounded by

$$\dim_{\mathrm{DE}}(\mathcal{P}_{h,i} - P_{h,i}^*, \Pi_h, 1/\sqrt{K}) \leq \tilde{O}(d)$$

for each $i \in [d_{\mathrm{s}}]$.

More generally, consider a general transition class with a linear structure

$$\mathcal{P}_h = \left\{ \mu_h(\cdot) \in \mathbb{R}^d : P_h(\cdot \mid s_h, a_h, e_h) = \phi^\top(s_h, a_h, e_h)\mu_h(\cdot) \right\},$$

where $\mu_h(\cdot)$ is a measure over $\mathcal{S}$. Then

$$\mathcal{P}_h \mathcal{G}_{h+1} = \left\{ (g, \mu_h(\cdot)) \in \mathcal{G}_{h+1} \times \mathbb{R}^d : P_h g_{h+1}(s_h, a_h, e_h) = \left\langle \phi(s_h, a_h, e_h), \int_s \mu_h(s)g(s)\mathrm{d}s \right\rangle \right\},$$

which is again a linear function class with respect to $\phi$. Therefore, we still have

$$\dim_{\mathrm{DE}}(\mathcal{P}_h \mathcal{G}_{h+1} - P_h^* \mathcal{G}_{h+1}, \Pi_h, 1/\sqrt{K}) \leq \tilde{O}(d)$$

if $\|\phi\|_2$ and $\|\int_s \mu_h(s)\mathrm{d}s\|_2$ are both uniformly upper bounded.

As a final remark, there are also other function classes whose distributional Eluder dimension is small, such as the generalized linear function class (Russo & Van Roy, 2013; Jin et al., 2021), the kernel function class with bounded effective dimension (Jin et al., 2021; Du et al., 2021). Since this is not the main point of the paper, we refer the readers to the mentioned papers for further details.

## F. The Nonparametric Instrumental Variable Model

We have provided a detailed explanation of the causal structure in the strategic MDP in Appendix C. In this section, we show that we can use the nonparametric instrumental variable (NPIV) model (see, e.g., Dikkala et al. (2020); Chen & Qi (2022)) to estimate the underlying model $R^*$ and $P^*$.

Recall the general form of the NPIV model (Chen & Qi, 2022)

$$Y = f^*(X) + U, \quad \text{with} \quad \mathbb{E}[U \mid W] = 0,$$

where $f^*$ is the unknown function to estimate, $Y$ is the response, $X$ is called endogenous variables, $W$ is called instrumental variables, and $U$ is the random (endogenous) noise.

According to the causal relationship (see Figure 2), the NPIV model is exactly applicable to estimate $P^*$ and $R^*$ with $Y = (r_h, s_{h+1})$, $X = (s_h, a_h, e_h)$, and $W = (s_h, a_h)$ since the noise $U$ is zero-mean in the population level (see, e.g., Equation (C.1)).

To solve for $f^*$, we build the conditional moment equation (Dikkala et al., 2020)

$$\mathbb{E}\left[Y - f^*(X) \mid W\right] = 0,$$

and construct an empirical dataset for $X, Y, W$ to perform least-square regression according to the conditional moment equation. Denote the least-square estimator of $f^*$ by $\hat{f}$, standard analysis allows us to bound the projected Mean Squared Error (pMSE)

$$\mathbb{E}_W\left[\mathbb{E}_{X,Y}\left[Y - \hat{f}(X) \mid W\right]^2\right].$$

Sometimes we care about the MSE of $\hat{f}$ under the population of $X, Y, W$, namely,

$$\mathbb{E}_{X,Y,W}\left[\left(Y - \hat{f}(X)\right)^2\right].$$

We can transfer pMSE to MSE via the ill-posed condition (Dikkala et al., 2020; Chen & Qi, 2022; Yu et al., 2022; Uehara et al., 2022a), which is standard in the literature of the NPIV model.

## G. Auxiliary Lemmas

In this section, we first introduce some lemmas that will be repeatedly used throughout the proof process.

**Lemma G.1** (Freedman's Inequality (see, e.g., Lemma 9 of Agarwal et al. (2014))). *Let $(X_t)_{t=1}^T$ be a real-valued martingale difference sequence adapted to the filtration $\mathcal{F}_t$. Let $\mathbb{E}_t \stackrel{\text{def}}{=} \mathbb{E}[\cdot \mid \mathcal{F}_t]$ be the conditional expectation. Suppose $|X_t| \leq R$ almost surely, then for any $0 < \lambda \leq 1/R$ it holds with probability at least $1 - \delta$ that*

$$\sum_{t=1}^T X_t \leq \lambda(e-2) \sum_{t=1}^T \mathbb{E}_{t-1}[X_t^2] + \frac{\log(1/\delta)}{\lambda}.$$

**Lemma G.2** (Rephrased from Theorem 1.3 of Devroye et al. (2018)). *For one-dimensional Gaussian distributions, we have*

$$\text{TV}\left(\mathcal{N}\left(\mu_1, \sigma_1^2\right), \mathcal{N}\left(\mu_2, \sigma_2^2\right)\right) \leq \frac{3\left|\sigma_1^2 - \sigma_2^2\right|}{2\sigma_1^2} + \frac{|\mu_1 - \mu_2|}{2\sigma_1},$$

*where* TV *denotes the total variation distance.*

**Lemma G.3** (Lemma 41 of Jin et al. (2021)). *Consider a function class $\Phi$ defined on $\mathcal{X}$ with $|\phi(x)| \leq C$ for all $\phi \in \Phi$ and $x \in \mathcal{X}$, and a distribution class $\Pi$ over $\mathcal{X}$. Suppose sequence $\{\phi_k\}_{k=1}^K \subset \Phi$ and $\{\mu_k\}_{k=1}^K \subset \Pi$ satisfy for all $k \in [K]$, $\sum_{t=1}^{k-1}(\mathbb{E}_{\mu_t}[\phi_k])^2 \leq \beta$. Then for all $k \in [K]$ and $\omega > 0$, it holds that*

$$\sum_{t=1}^k |\mathbb{E}_{\mu_t}[\phi_t]| \leq O\left(\sqrt{\dim_{\text{DE}}(\Phi, \Pi, \omega)\beta k} + \min\{k, \dim_{\text{DE}}(\Phi, \Pi, \omega)\}C + k\omega\right).$$

## H. Missing Proofs

We provide the formal proof for Theorem 5.4 in this section. Recall that $\bar{\mathcal{M}}^*$ is the aggregated model under the target distribution. Denoting the value function of any policy $\pi$ on $\bar{\mathcal{M}}^*$ as $\bar{V}_{\bar{\mathcal{M}}^*}^\pi$, we can use $\bar{\pi}^*$ to measure the regret of any algorithm $\mathcal{ALG}$ as

$$\text{Reg}(\mathcal{ALG}, K) \overset{\text{def}}{=} \sum_{k=1}^K \bar{V}_{\bar{\mathcal{M}}^*,1}^{\bar{\pi}^*}(s_1) - \bar{V}_{\bar{\mathcal{M}}^*,1}^{\pi^k}(s_1), \tag{H.1}$$

where $\pi^k$ is the policy committed by $\mathcal{ALG}$ in the $k$-th episode.

We assume $\mathcal{P}, \mathcal{R}, \mathcal{F}, \mathcal{G}$ are finite sets to simplify the analysis. Note that the capacity of the function space can be replaced by the corresponding covering number (Jin et al., 2021; Uehara et al., 2022b).

Before coming to the proofs, we prove some concentration lemmas at first. We define the norms of $f_h \in \mathcal{F}_h$ as

$$\left\|f_h^k\right\|_2^2 \overset{\text{def}}{=} \sum_{j=1}^k \mathbb{E}_{s_h, a_h \sim d_h^{s, \pi_j}}\left[f_h^2(s_h, a_h)\right] \tag{H.2}$$

$$\left\|f_h^k\right\|_{2,k}^2 \overset{\text{def}}{=} \sum_{j=1}^k f_h^2\left(s_h^j, a_h^j\right). \tag{H.3}$$

Let the filtration $\mathcal{F}_k$ be induced by $\{(s_1^j, a_1^j, e_1^j, r_1^j, ..., s_H^j, a_H^j, e_H^j, r_H^j)\}_{j=1}^k$. We can use the Freedman's inequality to bound the difference between them.

**Lemma H.1.** *With probability at least $1 - \delta/2$, for any $f = \{f_h\}_{h=1}^H$, any $(k, h) \in [K] \times [H]$, we have*

$$\left|\left\|f_h^k\right\|_{2,k}^2 - \left\|f_h^k\right\|_2^2\right| \leq \frac{1}{2}\left\|f_h^k\right\|_2^2 + 4(e-2)B^2\log(4KH|\mathcal{F}|/\delta).$$

*Proof.* Consider the martingale difference sequence $X_j \overset{\text{def}}{=} f_h^2(s_h^j, a_h^j) - \mathbb{E}_{s_h, a_h \sim d_h^{s, \pi_j}}[f_h^2(s_h, a_h)]$. Then $\mathbb{E}[X_j \mid \mathcal{F}_{j-1}] = 0$ since $\pi^j$ is $\mathcal{F}_{j-1}$-measurable. Moreover, $|X_j| \leq 2B^2$ almost surely.

Note that $\mathbb{E}[X_j^2 \mid \mathcal{F}_{j-1}] \leq B^2\mathbb{E}_{s_h, a_h \sim d_h^{s, \pi_j}}[f_h^2(s_h, a_h)]$, we invoke the Freedman's inequality (cf. Lemma G.1) with a union bound over $\mathcal{F}_h \times [K] \times [H]$ to show that with probability at least $1 - \delta/2$ for any $\lambda < 1/2B^2$,

$$\left|\left\|f_h^k\right\|_{2,k}^2 - \left\|f_h^k\right\|_2^2\right| \leq \lambda B^2(e-2)\left\|f_h^k\right\|_2^2 + \frac{\log(4KH|\mathcal{F}|/\delta)}{\lambda}, \forall f_h, k, h.$$

Choosing $\lambda = 1/4(e-2)B^2$ completes the proof. $\square$

For any $\nu_h \in \{\mathcal{R}_h - R_h^*, \mathcal{P}_{h,i} - G_{h,i}^*, (\mathcal{P}_h - P_h^*)\mathcal{G}_{h+1}\}$, assume that we have $\|\nu_h\|_\infty \le B$. Recall the definition of the risk function and empirical risk function (see, e.g., Equations (4.3) and (4.4)), namely,

$$l_h^k(\nu_h, f_h) = \sum_{j=1}^k \mathbb{E}_{s_h, a_h, e_h \sim d_h^{s, \pi^j}} \left[ f_h(s_h, a_h)\nu_h(s_h, a_h, e_h) \right],$$

$$\hat{l}_h^k(\nu_h, f_h) = \sum_{j=1}^k f_h(s_h^j, a_h^j)\hat{\nu}_h(s_h^j, a_h^j, e_h^j).$$

With a little abuse of notations, we use $\hat{\nu}_h(s_h^j, a_h^j, e_h^j)$ to denote a sample from $s_h, a_h, e_h \sim d_h^{s, \pi^j}$ on $\nu_h$, to wit,

- For $\nu_h \in \mathcal{R}_h - R_h^*$, $\hat{\nu}_h(s_h^j, a_h^j, e_h^j) \overset{\text{def}}{=} R_h(s_h^j, a_h^j, e_h^j) - r_h^j$.

- For $\nu_h \in \mathcal{P}_{h,i} - G_{h,i}^*$, $\hat{\nu}_h(s_h^j, a_h^j, e_h^j) \overset{\text{def}}{=} G_{h,i}(s_h^j, a_h^j, e_h^j) - s_{h+1,i}^j$.

- For $\nu_h \in (\mathcal{P}_h - P_h^*)\mathcal{G}_{h+1}$, $\hat{\nu}_h(s_h^j, a_h^j, e_h^j) \overset{\text{def}}{=} P_h g_{h+1}(s_h^j, a_h^j, e_h^j) - g_{h+1}(s_{h+1}^j)$.

For any $(k, h) \in [K] \times [H]$ and $\nu_h \in \mathcal{Y}_h$ for $\mathcal{Y}_h \in \{\mathcal{R}_h - R_h^*, \mathcal{P}_{h,i} - G_{h,i}^*, (\mathcal{P}_h - P_h^*)\mathcal{G}_{h+1}\}$, we have the following lemma using Freedman's inequality.

**Lemma H.2.** *With probability at least $1 - \delta/2$, for any $(k, h) \in [K] \times [H]$ and $\nu_h \in \mathcal{Y}_h, f_h \in \mathcal{F}_h$*

$$\left| l_h^k(\nu_h, f_h) - \hat{l}_h^k(\nu_h, f_h) \right| \le 4B\sqrt{\log(6KH|\mathcal{F}||\mathcal{Y}|/\delta)}\|f_h^k\|_2 + 4B^2 \log(6KH|\mathcal{F}||\mathcal{Y}|/\delta).$$

*Proof.* Let the martingale difference be

$$X_j \overset{\text{def}}{=} f_h(s_h^j, a_h^j)\hat{\nu}_h(s_h^j, a_h^j, e_h^j) - \mathbb{E}_{s_h, a_h, e_h \sim d_h^{s, \pi^j}}\left[ f_h(s_h, a_h)\nu_h(s_h, a_h, e_h) \right].$$

We know that $|X_j| \le 2B^2$ almost surely.

For a fixed $(k, h, \nu_h, f_h)$ tuple, observe that

$$\sum_{j=1}^k \mathbb{E}[X_j^2 \mid \mathcal{F}_{j-1}] \le \sum_{j=1}^k \mathbb{E}_{s_h, a_h, e_h \sim d_h^{s, \pi^j}}\left[ f_h^2(s_h, a_h)\nu_h^2(s_h, a_h, e_h) \right] \le B^2 \left\| f_h^k \right\|_2^2.$$

since $\pi^j$ is $\mathcal{F}_{j-1}$-measurable.

Therefore, if

$$\frac{\sqrt{\log(1/\delta)}}{B \left\| f_h^k \right\|_2 \sqrt{e-2}} > \frac{1}{2B^2}$$

holds, then

$$\left\| f_h^k \right\|_2^2 < \frac{4B^2 \log(1/\delta)}{e-2}.$$

Invoking Freedman's inequality (cf. Lemma G.1) by $\lambda = 1/2B^2$ yields that

$$\left| l_h^k(\nu_h, f_h) - \hat{l}_h^k(\nu_h, f_h) \right| \le \lambda(e-2)B^2 \left\| f_h^k \right\|_2^2 + \frac{\log(1/\delta)}{\lambda} \le 4B^2 \log(1/\delta).$$

Otherwise, we invoke Freedman's inequality with

$$\lambda = \frac{\sqrt{\log(1/\delta)}}{B \left\| f_h^k \right\|_2 \sqrt{e-2}} \le \frac{1}{2B^2}$$

yielding

$$\left| l_h^k(\nu_h, f_h) - \hat{l}_h^k(\nu_h, f_h) \right| \le 2B\sqrt{(e-2)\log(1/\delta)} \left\| f_h^k \right\|_2.$$

The lemma is finally proved by taking union bound over all $(k, h, \nu_h, f_h)$. $\qquad\square$

**Lemma H.3** (Consistent Confidence Set). *Define the confidence level $\beta_1, \beta_2, \beta_3$ in OPME by*

$$\begin{aligned}
\beta_1 &\overset{\text{def}}{=} 28B^2 \log(KH|\mathcal{F}||\mathcal{R}|/\delta) \\
\beta_2 &\overset{\text{def}}{=} 28B^2 \log(KH|\mathcal{F}||\mathcal{G}||\mathcal{P}|/\delta) \\
\beta_3 &\overset{\text{def}}{=} 28B^2 \log(KH|\mathcal{F}||\mathcal{P}|/\delta).
\end{aligned} \tag{H.4}$$

*With probability at least $1 - \delta$, the confidence set $\bar{\mathcal{C}}^k$ is consistent in that the underlying model is contained in $\mathcal{C}^k$, namely,*

$$R_h^* \in \mathcal{R}_h^k, G_{h,i}^* \in \mathcal{P}_{h,i}^k, P_h^* \in \mathcal{P}_h^k, \bar{\mathcal{M}}^* \in \bar{\mathcal{C}}^k, \quad \forall (k, h, i) \in [K] \times [H] \times [d_{\mathrm{s}}].$$

*Proof.* We assume the high probability events in Lemma H.1 and H.2 hold for simplicity. Note that the probability that either of them fails is at most $\delta$.

Lemma H.1 implies for any $f_h \in \mathcal{F}_h$

$$\frac{1}{2} \left\| f_h^k \right\|_2^2 - 4(e-2)B^2 \log(2KH|\mathcal{F}|/\delta) \le \left\| f_h^k \right\|_{2,k}^2 \le \frac{3}{2} \left\| f_h^k \right\|_2^2 + 4(e-2)B^2 \log(2KH|\mathcal{F}|/\delta)$$

Consider $\hat{L}_h^k(\iota_h^*) = \max_{f_h} \hat{l}_h^k(\iota_h^*, f_h) - \|f_h^k\|_{2,k}^2/2$ for $\iota_h^* \in \{R_h^*, G_{h,i}^*, P_h^* \mathcal{G}_{h+1}\}$. For any fixed $f_h \in \mathcal{F}_h$ we have

$$\begin{aligned}
&\hat{l}_h^k(\iota_h^*, f_h) - \frac{\|f_h^k\|_{2,k}^2}{2} \\
&\le l_h^k(\iota_h^*, f_h) - \frac{\|f_h^k\|_{2,k}^2}{2} + 4B\sqrt{\log(KH|\mathcal{F}||\mathcal{Y}|/\delta)}\|f_h^k\|_2 + 4B^2 \log(3KH|\mathcal{F}||\mathcal{Y}|/\delta) \\
&\le l_h^k(\iota_h^*, f_h) - \frac{\|f_h^k\|_2^2}{4} + 4B\sqrt{\log(KH|\mathcal{F}||\mathcal{Y}|/\delta)}\|f_h^k\|_2 \\
&\quad + 4B^2 \log(3KH|\mathcal{F}||\mathcal{Y}|/\delta) + 2(e-2)B^2 \log(2KH|\mathcal{F}|/\delta) \\
&\le -\frac{\|f_h^k\|_2^2}{4} + 4B\sqrt{\log(KH|\mathcal{F}||\mathcal{Y}|/\delta)}\|f_h^k\|_2 \\
&\quad + 4B^2 \log(3KH|\mathcal{F}||\mathcal{Y}|/\delta) + 2(e-2)B^2 \log(2KH|\mathcal{F}|/\delta).
\end{aligned}$$

Here $\mathcal{Y}$ denotes the corresponding function space $\mathcal{R}_h - R_h^*$, $\mathcal{P}_{h,i} - G_{h,i}^*$, or $(\mathcal{P}_h - P_h^*)\mathcal{G}_{h+1}$. The first inequality is by Lemma H.2. The second inequality is due to Lemma H.1. The third inequality holds because $l_h^k(\iota_h^*, f_h) = 0$ for any $f_h$ by definition (see, e.g., Equation (4.3)) .

Note that Equation (H.5) is a standard quadratic function with respect to $\|f_h^k\|_2$, we have that

$$\begin{aligned}
&\hat{l}_h^k(\iota_h^*, f_h) - \frac{\|f_h^k\|_{2,k}^2}{2} \\
&\le -\frac{\|f_h^k\|_2^2}{4} + 4B\sqrt{\log(KH|\mathcal{F}||\mathcal{Y}|/\delta)}\|f_h^k\|_2 + 4B^2 \log(3KH|\mathcal{F}||\mathcal{Y}|/\delta) + 2(e-2)B^2 \log(2KH|\mathcal{F}|/\delta) \\
&\le 28B^2 \log(KH|\mathcal{F}||\mathcal{Y}|/\delta)
\end{aligned}$$

holds for any $f_h \in \mathcal{F}_h$.

Therefore, with probability at least $1 - \delta$, for any $(k, h) \in [K] \times [H]$, it holds that

$$\hat{L}_h^k(\iota_h^*) = \max_{f_h} \hat{l}_h^k(\iota_h^*, f_h) - \frac{\|f_h^k\|_{2,k}^2}{2} \le 28B^2 \log(KH|\mathcal{F}||\mathcal{Y}|/\delta).$$

By the definition of $\beta_1, \beta_2, \beta_3$, we can finish the proof. $\qquad\square$

**Lemma H.4.** *With probability at least $1 - \delta$, it holds the following statements.*

*For any $(k, h, i) \in [K] \times [H] \times [d_\mathrm{s}]$ and $R_h \in \mathcal{R}_h^k, G_{h,i} \in \mathcal{P}_{h,i}^k$,*

$$\sum_{j=1}^k \mathbb{E}_{(s_h, a_h) \sim d_h^{\mathrm{s}, \pi^j}} \left[ \left( \mathbb{E}_{e_h \sim F_h(\cdot | s_h, a_h, \mathcal{P}_h^\mathrm{s})} \left[ R_h(s_h, a_h, e_h) - R_h^*(s_h, a_h, e_h) \right] \right)^2 \right] = O(\beta_1),$$

$$\sum_{j=1}^k \mathbb{E}_{(s_h, a_h) \sim d_h^{\mathrm{s}, \pi^j}} \left[ \left( \mathbb{E}_{e_h \sim F_h(\cdot | s_h, a_h, \mathcal{P}_h^\mathrm{s})} \left[ G_{h,i}(s_h, a_h, e_h) - G_{h,i}^*(s_h, a_h, e_h) \right] \right)^2 \right] = O(\beta_3).$$

*For any $(k, h) \in [K] \times [H]$ and $P_h \in \mathcal{P}_h^k, g_{h+1} \in \mathcal{G}_{h+1}$,*

$$\sum_{j=1}^k \mathbb{E}_{(s_h, a_h) \sim d_h^{\mathrm{s}, \pi^j}} \left[ \left( \mathbb{E}_{e_h \sim F_h(\cdot | s_h, a_h, \mathcal{P}_h^\mathrm{s})} \left[ P_h g_{h+1}(s_h, a_h, e_h) - P_h^* g_{h+1}(s_h, a_h, e_h) \right] \right)^2 \right] = O(\beta_2).$$

*Proof.* We assume the high probability events in Lemma H.1 and H.2 happen for simplicity. Note that the probability either of them fails is at most $\delta$.

For any $\nu_h \in \{\mathcal{R}_h^k - R_h^*, \mathcal{P}_{h,i}^k - G_{h,i}^*, (\mathcal{P}_h^k - P_h^*)\mathcal{G}_{h+1}\}$, define function $f[\nu_h]$ as

$$f[\nu_h] : \mathcal{S} \times \mathcal{A} \to \mathbb{R} \text{ such that } f[\nu_h](s_h, a_h) = \mathbb{E}_{e_h \sim F_h(\cdot | s_h, a_h, \mathcal{P}_h^\mathrm{s})} \left[ \nu_h(s_h, a_h, e) \right]. \tag{H.5}$$

By Assumption 5.1 we know $f[\nu_h] \in \mathcal{F}_h$ for any $\nu_h$.

By definition, we know

$$\sum_{j=1}^k \mathbb{E}_{(s_h, a_h) \sim d_h^{\mathrm{s}, \pi^j}} \left[ \left( \mathbb{E}_{e_h \sim F_h(\cdot | s_h, a_h, \mathcal{P}_h^\mathrm{s})} \left[ \nu_h(s_h, a_h, e_h) \right] \right)^2 \right] = \left\| f[\nu_h]^k \right\|_2^2 = l_h^k(\nu_h, f[\nu_h]). \tag{H.6}$$

The second equation holds because

$$l_h^k(\nu_h, f[\nu_h]) = \sum_{j=1}^k \mathbb{E}_{(s_h, a_h, e_h) \sim d_h^{\mathrm{s}, \pi^j}} \left[ f[\nu_h](s_h, a_h) \nu_h(s_h, a_h, e_h) \right]$$

$$= \sum_{j=1}^k \mathbb{E}_{(s_h, a_h) \sim d_h^{\mathrm{s}, \pi^j}} \left[ \mathbb{E}_{e_h \sim F_h(\cdot | s_h, a_h, \mathcal{P}_h^\mathrm{s})} \left[ f[\nu_h](s_h, a_h) \nu_h(s_h, a_h, e_h) \mid s_h, a_h \right] \right]$$

$$= \sum_{j=1}^k \mathbb{E}_{(s_h, a_h) \sim d_h^{\mathrm{s}, \pi^j}} \left[ f[\nu_h](s_h, a_h) \cdot \mathbb{E}_{e_h \sim F_h(\cdot | s_h, a_h, \mathcal{P}_h^\mathrm{s})} \left[ \nu_h(s_h, a_h, e_h) \mid s_h, a_h \right] \right]$$

$$= \sum_{j=1}^k \mathbb{E}_{(s_h, a_h) \sim d_h^{\mathrm{s}, \pi^j}} \left[ \left( \mathbb{E}_{e_h \sim F_h(\cdot | s_h, a_h, \mathcal{P}_h^\mathrm{s})} \left[ \nu_h(s_h, a_h, e_h) \right] \right)^2 \right].$$

For any $\nu_h \in \mathcal{Y}_h = \{\mathcal{R}_h^k - R_h^*, \mathcal{P}_{h,i}^k - G_{h,i}^*, (\mathcal{P}_h^k - P_h^*)\mathcal{G}_{h+1}\}$ and any $f_h$ we have

$$l_h^k(\nu_h, f_h) \leq \hat{l}_h^k(\nu_h, f_h) + 4B\sqrt{\log(3KH|\mathcal{F}||\mathcal{Y}|/\delta)} \|f_h^k\|_2 + 4B^2 \log(3KH|\mathcal{F}||\mathcal{Y}|/\delta)$$

$$\leq \beta + \frac{\|f_h^k\|_{2,k}^2}{2} + 4B\sqrt{\log(3KH|\mathcal{F}||\mathcal{Y}|/\delta)} \|f_h^k\|_2 + 4B^2 \log(3KH|\mathcal{F}||\mathcal{Y}|/\delta)$$

$$\leq \beta + \frac{3\|f_h^k\|_2^2}{4} + 4B\sqrt{\log(3KH|\mathcal{F}||\mathcal{Y}|/\delta)} \|f_h^k\|_2 + 4B^2 \log(3KH|\mathcal{F}||\mathcal{Y}|/\delta) + 2(e-2)B^2 \log(2KH|\mathcal{F}|/\delta).$$

Here $\beta \in \{\beta_1, \beta_3, \beta_2\}$ is the corresponding confidence level of $\mathcal{Y}_h$. The first inequality is by Lemma H.2. The second inequality is due to the construction of the confidence sets $\mathcal{R}_h^k, \mathcal{P}_{h,i}^k, \mathcal{P}_h^k$ according to Equations (4.7), (4.8) and (D.5). The third inequality is by Lemma H.1.

Plugging in $f_h = f[\nu_h]$ implies

$$
\begin{aligned}
l_h^k(\nu_h, f[\nu_h]) &= \left\| f[\nu_h]^k \right\|_2^2 \\
&\leq \beta + \frac{3 \left\| f[\nu_h]^k \right\|_2^2}{4} + 4B\sqrt{\log(3KH|\mathcal{F}||\mathcal{Y}|/\delta)} \cdot \| f[\nu_h]^k \|_2 \\
&\quad + 4B^2 \log(3KH|\mathcal{F}||\mathcal{Y}|/\delta) + 2(e-2)B^2 \log(2KH|\mathcal{F}|/\delta).
\end{aligned}
$$

Thus we have

$$
\begin{aligned}
\frac{\left\| f[\nu_h]^k \right\|_2^2}{4} &\leq \beta + 4B\sqrt{\log(3KH|\mathcal{F}||\mathcal{Y}|/\delta)} \| f[\nu_h]^k \|_2 + 4B^2 \log(3KH|\mathcal{F}||\mathcal{Y}|/\delta) \\
&\quad + 2(e-2)B^2 \log(2KH|\mathcal{F}|/\delta).
\end{aligned}
$$

Solving this expression proves the result. $\qquad\square$

Now we use the ill-posedness measure (Definition 5.2) and knowledge transfer multiplicative term (Definition 5.3) to propose the following lemma.

**Lemma H.5.** *With probability at least $1 - \delta$ it holds the following statements.*

*For any $(k, h, i) \in [K] \times [H] \times [d_s]$ and $R_h \in \mathcal{R}_h^k, G_{h,i} \in \mathcal{P}_{h,i}^k$,*

$$
\sum_{j=1}^{k} \mathbb{E}_{(s_h, a_h, e_h) \sim d_h^{t, \pi^j}} \left[ (R_h(s_h, a_h, e_h) - R_h^*(s_h, a_h, e_h))^2 \right] = O\left( \beta_1 \tau_h C_h^f \right),
$$

$$
\sum_{j=1}^{k} \mathbb{E}_{(s_h, a_h, e_h) \sim d_h^{t, \pi^j}} \left[ (G_{h,i}(s_h, a_h, e_h) - G_{h,i}^*(s_h, a_h, e_h))^2 \right] = O\left( \beta_3 \tau_h C_h^f \right).
$$

*For any $(k, h) \in [K] \times [H]$ and $P_h \in \mathcal{P}_h^k, g_{h+1} \in \mathcal{G}_{h+1}$,*

$$
\sum_{j=1}^{k} \mathbb{E}_{(s_h, a_h, e_h) \sim d_h^{t, \pi^j}} \left[ (P_h g_{h+1}(s_h, a_h, e_h) - P_h^* g_{h+1}(s_h, a_h, e_h))^2 \right] = O\left( \beta_2 \tau_h C_h^f \right).
$$

*Proof.* This is a straightforward application of Lemma H.4.

Take the reward function as an example, we have

$$
\sum_{j=1}^{k} \mathbb{E}_{(s_h, a_h) \sim d_h^{s, \pi^j}} \left[ \left( \mathbb{E}_{e_h \sim F_h(\cdot|s_h, a_h, \mathcal{P}_h^s)} [R_h(s_h, a_h, e_h) - R_h^*(s_h, a_h, e_h)] \right)^2 \right] = O\left( \beta_1 \right) \tag{H.7}
$$

according to Lemma H.4.

By Equation (H.7) and the definition of $\tau_h$, it holds that

$$
\sum_{j=1}^{k} \mathbb{E}_{(s_h, a_h, e_h) \sim d_h^{s, \pi^j}} \left[ (R_h(s_h, a_h, e_h) - R_h^*(s_h, a_h, e_h))^2 \right] = O\left( \beta_1 \tau_h \right). \tag{H.8}
$$

By Equation (H.8) and the definition of $C_h^f$ we have

$$
\sum_{j=1}^{k} \mathbb{E}_{(s_h, a_h, e_h) \sim d_h^{t, \pi^j}} \left[ (R_h(s_h, a_h, e_h) - R_h^*(s_h, a_h, e_h))^2 \right] = O\left( \beta_1 \tau_h C_h^f \right).
$$

The proof is identical for the rest two terms. $\qquad\square$

**Lemma H.6** (Regret Decomposition)**.** *Let $d^{\pi}_{\bar{\mathcal{M}}^*,h}(s,a) \stackrel{\text{def}}{=} \Pr^{\pi}_{\bar{\mathcal{M}}^*}(s_h = s, a_h = a)$ be the occupancy measure of $\pi$ under the unknown aggregated model $\bar{\mathcal{M}}^*$. Recall that the optimistic model $\bar{\mathcal{M}}^k = \bar{\mathcal{M}}(R^k, P^k)$ is defined in OPME (Algorithm 2). The regret can be decomposed as*

$$\text{Reg}(\textit{OPME-D}, K) = \sum_{k=1}^{K} \bar{V}^{\bar{\pi}^*}_{\mathcal{M}^*,1}(s_1) - \bar{V}^{\pi^k}_{\mathcal{M}^*,1}(s_1)$$

$$\lesssim \sum_{k=1}^{K} \sum_{h=1}^{H} \mathbb{E}_{(s_h,a_h,e_h)\sim d^{\mathrm{t},\pi^k}_h}\Bigg[ \Big| R^k_h(s_h, a_h, e_h) - R^*_h(s_h, a_h, e_h)\Big|$$

$$+ H \sum_{i=1}^{d_{\mathrm{s}}} \Big| G^k_{h,i}(s_h, a_h, e_h) - G^*_{h,i}(s_h, a_h, e_h)\Big| \Bigg]$$

*for the dynamical system transition class, or it can be decomposed as*

$$\text{Reg}(\textit{OPME-G}, K) = \sum_{k=1}^{K} \bar{V}^{\bar{\pi}^*}_{\mathcal{M}^*,1}(s_1) - \bar{V}^{\pi^k}_{\bar{\mathcal{M}}^*,1}(s_1)$$

$$\lesssim \sum_{k=1}^{K} \sum_{h=1}^{H} \mathbb{E}_{(s_h,a_h,e_h)\sim d^{\mathrm{t},\pi^k}_h}\Big[ \Big| R^k_h(s_h, a_h, e_h) - R^*_h(s_h, a_h, e_h)\Big| \Big]$$

$$+ \sum_{k=1}^{K} \sum_{h=1}^{H} \mathbb{E}_{(s_h,a_h,e_h)\sim d^{\mathrm{t},\pi^k}_h}\Big[ \Big| P^k_h \bar{V}^{\pi^k}_{\mathcal{M}^k,h+1}(s_h, a_h, e_h) - P^*_h \bar{V}^{\pi^k}_{\mathcal{M}^k,h+1}(s_h, a_h, e_h)\Big| \Big]$$

*for the general transition class.*

*Proof.* The regret can be decomposed as

$$\text{Reg}(\text{OPME-D}, K) = \sum_{k=1}^{K} \bar{V}^{\bar{\pi}^*}_{\mathcal{M}^*,1}(s_1) - \bar{V}^{\pi^k}_{\bar{\mathcal{M}}^*,1}(s_1) \leq \sum_{k=1}^{K} \bar{V}^{\pi^k}_{\bar{\mathcal{M}}^k,1}(s_1) - \bar{V}^{\pi^k}_{\bar{\mathcal{M}}^*,1}(s_1)$$

$$\leq \sum_{k=1}^{K} \sum_{h=1}^{H} \mathbb{E}_{(s_h,a_h)\sim d^{\pi^k}_{\bar{\mathcal{M}}^*,h}}\Big[ \Big| \bar{R}^k_h(s_h, a_h) - \bar{R}^*_h(s_h, a_h)\Big| + \Big| \bar{P}^k_h \bar{V}^{\pi^k}_{\bar{\mathcal{M}}^k,h+1}(s_h, a_h) - \bar{P}^*_h \bar{V}^{\pi^k}_{\bar{\mathcal{M}}^k,h+1}(s_h, a_h)\Big| \Big]$$

$$\leq \sum_{k=1}^{K} \sum_{h=1}^{H} \mathbb{E}_{(s_h,a_h)\sim d^{\pi^k}_{\bar{\mathcal{M}}^*,h}}\Big[ \Big| \bar{R}^k_h(s_h, a_h) - \bar{R}^*_h(s_h, a_h)\Big| \Big]$$

$$+ H \sum_{k=1}^{K} \sum_{h=1}^{H} \mathbb{E}_{(s_h,a_h)\sim d^{\pi^k}_{\bar{\mathcal{M}}^*,h}}\Big[ \Big\| \bar{P}^k_h(\cdot \mid s_h, a_h) - \bar{P}^*_h(\cdot \mid s_h, a_h)\Big\|_1 \Big].$$

For the transition term, we can bound it by

$$\sum_{k=1}^{K} \sum_{h=1}^{H} \mathbb{E}_{(s_h,a_h)\sim d^{\pi^k}_{\bar{\mathcal{M}}^*,h}}\Big[ \Big\| \bar{P}^k_h(\cdot \mid s_h, a_h) - \bar{P}^*_h(\cdot \mid s_h, a_h)\Big\|_1 \Big]$$

$$= \sum_{k=1}^{K} \sum_{h=1}^{H} \mathbb{E}_{(s_h,a_h)\sim d^{\pi^k}_{\bar{\mathcal{M}}^*,h}}\Big[ \Big\| \mathbb{E}_{t_h\sim\mathcal{P}^{\mathrm{t}}_h, e_h\sim F_h(\cdot|s_h,a_h,t_h)} \big[ P^k_h(\cdot \mid s_h, a_h, e_h, t_h) - P^*_h(\cdot \mid s_h, a_h, e_h, t_h)\big]\Big\|_1 \Big]$$

$$\leq \sum_{k=1}^{K} \sum_{h=1}^{H} \mathbb{E}_{(s_h,a_h)\sim d^{\pi^k}_{\bar{\mathcal{M}}^*,h}, t_h\sim\mathcal{P}^{\mathrm{t}}_h, e_h\sim F_h(\cdot|s_h,a_h,t_h)}\Big[ \Big\| P^k_h(\cdot \mid s_h, a_h, e_h, t_h) - P^*_h(\cdot \mid s_h, a_h, e_h, t_h)\Big\|_1 \Big].$$

The first inequality uses Jensen's inequality. Let $F_h(\cdot \mid s_h, a_h, \mathcal{P}^{\mathrm{t}}_h)$ be the distribution of $e_h$ under $s_h, a_h$ and $t_h \sim \mathcal{P}^{\mathrm{t}}_h$. For the dynamical system transition class, we know $P_h(\cdot \mid s_h, a_h, e_h, t_h)$ is a $d_{\mathrm{s}}$-dimensional Gaussian distribution with mean

$G_h(s_h, a_h, e_h)$ and covariance $I_{d_s}$ (i.e., $d_s$ independent Gaussian random variables) by Equation (D.1). Therefore, we have

$$\sum_{k=1}^{K}\sum_{h=1}^{H} \mathbb{E}_{(s_h,a_h)\sim d_{\bar{\mathcal{M}}^*,h}^{\pi^k}, t_h\sim\mathcal{P}_h^t, e_h\sim F_h(\cdot|s_h,a_h,t_h)}\left[\left\|P_h^k(\cdot\mid s_h,a_h,e_h,t_h)-P_h^*(\cdot\mid s_h,a_h,e_h,t_h)\right\|_1\right]$$

$$\lesssim \sum_{k=1}^{K}\sum_{h=1}^{H} \mathbb{E}_{(s_h,a_h)\sim d_{\bar{\mathcal{M}}^*,h}^{\pi^k}, e_h\sim F_h(\cdot|s_h,a_h,\mathcal{P}_h^t)}\left[\left\|\boldsymbol{G}_h^k(s_h,a_h,e_h)-\boldsymbol{G}_h^*(s_h,a_h,e_h)\right\|_1\right],$$

by Lemma G.2.

This term can be further bounded by

$$\sum_{k=1}^{K}\sum_{h=1}^{H} \mathbb{E}_{(s_h,a_h)\sim d_{\bar{\mathcal{M}}^*,h}^{\pi^k}, e_h\sim F_h(\cdot|s_h,a_h,\mathcal{P}_h^t)}\left[\left\|\boldsymbol{G}_h^k(s_h,a_h,e_h)-\boldsymbol{G}_h^*(s_h,a_h,e_h)\right\|_1\right]$$

$$=\sum_{k=1}^{K}\sum_{h=1}^{H} \mathbb{E}_{(s_h,a_h,e_h)\sim d_h^{t,\pi^k}}\left[\left\|\boldsymbol{G}_h^k(s_h,a_h,e_h)-\boldsymbol{G}_h^*(s_h,a_h,e_h)\right\|_1\right]$$

$$=\sum_{k=1}^{K}\sum_{h=1}^{H}\sum_{i=1}^{d_s} \mathbb{E}_{(s_h,a_h,e_h)\sim d_h^{t,\pi^k}}\left[\left|G_{h,i}^k(s_h,a_h,e_h)-G_{h,i}^*(s_h,a_h,e_h)\right|\right]$$

where the first equality is by the definition of $d_h^{t,\pi^k}$ (see Section 3.3) and the aggregated model $\bar{\mathcal{M}}^*$.

Similarly, the reward term can be bounded by

$$\sum_{k=1}^{K}\sum_{h=1}^{H} \mathbb{E}_{(s_h,a_h)\sim d_{\bar{\mathcal{M}}^*,h}^{\pi^k}}\left[\left|\bar{R}_h^k(s_h,a_h)-\bar{R}_h^*(s_h,a_h)\right|\right]$$

$$\leq \sum_{k=1}^{K}\sum_{h=1}^{H} \mathbb{E}_{(s_h,a_h)\sim d_{\bar{\mathcal{M}}^*,h}^{\pi^k}, e_h\sim F_h(\cdot|s_h,a_h,\mathcal{P}_h^t)}\left[\left|R_h^k(s_h,a_h,e_h)-R_h^*(s_h,a_h,e_h)\right|\right]$$

$$=\sum_{k=1}^{K}\sum_{h=1}^{H} \mathbb{E}_{(s_h,a_h,e_h)\sim d_h^{t,\pi^k}}\left[\left|R_h^k(s_h,a_h,e_h)-R_h^*(s_h,a_h,e_h)\right|\right].$$

For the general transition class, we decompose the regret as

$$\text{Reg}(\text{OPME-G}, K) = \sum_{k=1}^{K}\bar{V}_{\bar{\mathcal{M}}^*,1}^{\bar{\pi}^*}(s_1)-\bar{V}_{\bar{\mathcal{M}}^*,1}^{\pi^k}(s_1) \leq \sum_{k=1}^{K}\bar{V}_{\bar{\mathcal{M}}^k,1}^{\pi^k}(s_1)-\bar{V}_{\bar{\mathcal{M}}^*,1}^{\pi^k}(s_1)$$

$$\leq \sum_{k=1}^{K}\sum_{h=1}^{H} \mathbb{E}_{(s_h,a_h)\sim d_{\bar{\mathcal{M}}^*,h}^{\pi^k}}\left[\left|\bar{R}_h^k(s_h,a_h)-\bar{R}_h^*(s_h,a_h)\right|+\left|\bar{P}_h^k\bar{V}_{\bar{\mathcal{M}}^k,h+1}^{\pi^k}(s_h,a_h)-\bar{P}_h^*\bar{V}_{\bar{\mathcal{M}}^k,h+1}^{\pi^k}(s_h,a_h)\right|\right].$$

The transition term now can be bounded by

$$\sum_{k=1}^{K}\sum_{h=1}^{H} \mathbb{E}_{(s_h,a_h)\sim d_{\bar{\mathcal{M}}^*,h}^{\pi^k}}\left[\left|\bar{P}_h^k\bar{V}_{\bar{\mathcal{M}}^k,h+1}^{\pi^k}(s_h,a_h)-\bar{P}_h^*\bar{V}_{\bar{\mathcal{M}}^k,h+1}^{\pi^k}(s_h,a_h)\right|\right]$$

$$=\sum_{k=1}^{K}\sum_{h=1}^{H} \mathbb{E}_{(s_h,a_h)\sim d_{\bar{\mathcal{M}}^*,h}^{\pi^k}}\left[\left|\mathbb{E}_{e_h\sim F_h(\cdot|s_h,a_h,\mathcal{P}_h^t)}\left[P_h^k\bar{V}_{\bar{\mathcal{M}}^k,h+1}^{\pi^k}(s_h,a_h,e_h)-P_h^*\bar{V}_{\bar{\mathcal{M}}^k,h+1}^{\pi^k}(s_h,a_h,e_h)\right]\right|\right]$$

$$\leq \sum_{k=1}^{K}\sum_{h=1}^{H} \mathbb{E}_{(s_h,a_h,e_h)\sim d_h^{t,\pi^k}}\left[\left|P_h^k\bar{V}_{\bar{\mathcal{M}}^k,h+1}^{\pi^k}(s_h,a_h,e_h)-P_h^*\bar{V}_{\bar{\mathcal{M}}^k,h+1}^{\pi^k}(s_h,a_h,e_h)\right|\right].$$

The last inequality is by Jensen's inequality. $\qquad\square$

Now we come to bound the regret of OPME.

**Theorem H.7.** *Under Assumption 5.1, with probability at least $1 - \delta$ the regret of OPME is bounded by*

$$\text{Reg}(\textit{OPME-D}, K) = \tilde{O}\left(\sum_{h=1}^{H} HB\sqrt{d_{\text{M},h}\tau_h C_h^{\text{f}} \log(|\mathcal{R}||\mathcal{P}||\mathcal{F}|/\delta)K}\right)$$

*for OPME-D, and*

$$\text{Reg}(\textit{OPME-G}, K) = \tilde{O}\left(\sum_{h=1}^{H} B\sqrt{d_{\text{V},h}\tau_h C_h^{\text{f}} \log(|\mathcal{R}||\mathcal{P}||\mathcal{G}||\mathcal{F}|/\delta)K}\right)$$

*for OPME-G.*

*Proof.* Fix $h \in [H]$. Since $R_h^k \in \mathcal{R}_h^{k-1}$ for any $k \in [K]$, we use Lemma G.3 with

$$C = B, \omega = \frac{1}{\sqrt{K}}, \Phi = \mathcal{R}_h - R_h^*, \phi_k = R_h^k - R_h^*, \mu_k = d_h^{\text{t},\pi^k}$$

to obtain

$$\sum_{k=1}^{K} \mathbb{E}_{(s_h,a_h,e_h)\sim d_h^{\text{t},\pi^k}}\left[\left|R_h^k(s_h,a_h,e_h) - R_h^*(s_h,a_h,e_h)\right|\right]$$

$$\leq O\left(\sqrt{\dim_{\text{DE}}(\mathcal{R}_h - R_h^*, \Pi_h, 1/\sqrt{K})\beta_1\tau_h C_h^{\text{f}} K}\right)$$

by Lemma H.5.

Similarly, for any $i \in [d_{\text{s}}]$, we have

$$\sum_{k=1}^{K} \mathbb{E}_{(s_h,a_h,e_h)\sim d_h^{\text{t},\pi^k}}\left[\left|G_{h,i}^k(s_h,a_h,e_h) - G_{h,i}^*(s_h,a_h,e_h)\right|\right]$$

$$\leq O\left(\sqrt{\dim_{\text{DE}}(\mathcal{P}_{h,i} - P_{h,i}^*, \Pi_h, 1/\sqrt{K})\beta_3\tau_h C_h^{\text{f}} K}\right).$$

By Lemma H.6, with probability at least $1 - \delta$, we can bound the regret of OPME-D as

$$\text{Reg}(\text{OPME-D}, K) \leq H\sum_{k=1}^{K}\sum_{h=1}^{H}\sum_{i=1}^{d_{\text{s}}} \mathbb{E}_{(s_h,a_h,e_h)\sim d_h^{\text{t},\pi^k}}\left[\left|G_{h,i}^k(s_h,a_h,e_h) - G_{h,i}^*(s_h,a_h,e_h)\right|\right]$$

$$+ \sum_{k=1}^{K}\sum_{h=1}^{H} \mathbb{E}_{(s_h,a_h,e_h)\sim d_h^{\text{t},\pi^k}}\left[\left|R_h^k(s_h,a_h,e_h) - R_h^*(s_h,a_h,e_h)\right|\right]$$

$$\leq \tilde{O}\left(\sum_{h=1}^{H} HB\sqrt{d_{\text{M},h}\tau_h C_h^{\text{f}} \log(|\mathcal{R}||\mathcal{P}||\mathcal{F}|/\delta)K}\right).$$

Similarly, we invoke the Lemma G.3 with

$$C = B, \omega = \frac{1}{\sqrt{K}}, \Phi = \mathcal{P}_h\mathcal{G}_{h+1} - P_h^*\mathcal{G}_{h+1}, \phi_k = P_h^k\bar{V}_{\mathcal{M}^k,h+1}^{\pi^k} - P_h^*\bar{V}_{\mathcal{M}^k,h+1}^{\pi^k}, \mu_k = d_h^{\text{t},\pi^k}$$

to obtain

$$\sum_{k=1}^{K}\sum_{h=1}^{H} \mathbb{E}_{(s_h,a_h,e_h)\sim d_h^{\text{t},\pi^k}}\left[\left|P_h^k\bar{V}_{\mathcal{M}^k,h+1}^{\pi^k}(s_h,a_h,e_h) - P_h^*\bar{V}_{\mathcal{M}^k,h+1}^{\pi^k}(s_h,a_h,e_h)\right|\right]$$

$$\leq O\left(\sqrt{\dim_{\text{DE}}(\mathcal{P}_h\mathcal{G}_{h+1} - P_h^*\mathcal{G}_{h+1}, \Pi_h, 1/\sqrt{K})\beta_2\tau_h C_h^{\text{f}} K}\right).$$

The regret of OPME-G is bounded as

$$\mathrm{Reg}\,(\text{OPME-G}, K) \leq \mathbb{E}_{(s_h, a_h, e_h) \sim d_h^{t, \pi^k}} \left[ \left| P_h^k \bar{V}_{\mathcal{M}^k, h+1}^{\pi^k}(s_h, a_h, e_h) - P_h^* \bar{V}_{\mathcal{M}^k, h+1}^{\pi^k}(s_h, a_h, e_h) \right| \right]$$

$$+ \sum_{k=1}^{K} \sum_{h=1}^{H} \mathbb{E}_{(s_h, a_h, e_h) \sim d_h^{t, \pi^k}} \left[ \left| R_h^k(s_h, a_h, e_h) - R_h^*(s_h, a_h, e_h) \right| \right]$$

$$\leq \tilde{O}\left( \sum_{h=1}^{H} B \sqrt{d_{\mathrm{V}, h} \tau_h C_h^{\mathrm{f}} \log(|\mathcal{R}||\mathcal{P}||\mathcal{G}||\mathcal{F}|/\delta) K} \right).$$

$\square$

*Proof of Theorem 5.4.* The sample complexity of OPME (Theorem 5.4) can be obtained by standard online-to-batch conversion (Jin et al., 2018) from Theorem H.7. That, the suboptimality of a uniform policy from $\{\pi_1, \pi_2, ..., \pi_K\}$ for a given number of episodes $K$ is at most

$$\tilde{O}\left( \frac{\sum_{h=1}^{H} HB\sqrt{d_{\mathrm{M}, h} \tau_h C_h^{\mathrm{f}} \log(|\mathcal{R}||\mathcal{P}||\mathcal{F}|/\delta)}}{\sqrt{K}} \right)$$

for OPME-D, and

$$\tilde{O}\left( \frac{\sum_{h=1}^{H} B\sqrt{d_{\mathrm{V}, h} \tau_h C_h^{\mathrm{f}} \log(|\mathcal{R}||\mathcal{P}||\mathcal{G}||\mathcal{F}|/\delta)}}{\sqrt{K}} \right)$$

for OPME-G by Theorem H.7. To bound this suboptimality term to be at most $\epsilon$, we can prove the theorem by setting $K$ according to the theorem. $\square$

## I. The Necessity of the Ill-posed Measure and Knowledge Transfer Multiplicative Term

Recall that the sample complexity of the model-based algorithm OPME-G has a linear dependency on the ill-posed measure $\tau_h$, the knowledge transfer multiplicative term $C_h^{\mathrm{f}}$, and the distributional Eluder dimension $d_{\mathrm{V}, h}$. Generally speaking, they are all necessary terms in our bound in order to find near-optimal policies. Now, we discuss the necessity of these terms.

- The distributional Eluder dimension is standard in the literature of RL with general function approximation (Jin et al., 2021). Without bounded distributional Eluder dimension, the sample complexity can scale linearly with the state space and action space in the worst case.

- The ill-posedness measure is also standard in the ill-posed inverse problems. This term is necessary as long as we hope to identify the underlying reward function $R^*$ and transition function $P^*$. Taking the reward function as an example, the mean square error of identification is lower bounded by the inverse of eigenvalues of an operator $K$, where $K$ is defined on $\mathcal{R}_h - R_h^*$ and $(K\nu_h)(s_h, a_h) := \mathbb{E}_{e_h}[\nu_h(s_h, a_h, e_h)]$, as pointed out by Chen & Reiss (2011); Hall & Horowitz (2005). In other words, $K$ is a linear mapping from the function space of $(s_h, a_h, e_h)$ (endogenous variables) to the function space of $(s_h, a_h)$ (instrumental variables). Suppose the eigenvalues of $K$ are $\lambda_1 \geq \lambda_2 \geq \cdots \geq 0$ and $\lambda_k \geq k^{-\alpha}$, then the lower bound of the MSE scales as $\Omega(n^{-\beta/(\beta+\alpha)})$ ($n$ is the number of samples) for $\alpha, \beta \geq 0$. Here $\alpha$ controls the magnitude of the ill-posedness measure defined in the paper, and if $\alpha \to \infty$ the estimation error will diverge even if $n$ is very large.

- The third term is the distributional shift term. Our problem is similar to the covariate shift setting in the unsupervised domain adaptation. We can regard the model as using $s_h, a_h, e_h$ to predict $r_h, s_{h+1}$ with underlying functions $R_h^*, P_h^*$. The source distribution is $d_h^{\pi, s}$ and the target distribution is $d_h^{\pi, t}$ for an arbitrary policy $\pi$. Therefore, a well-known conditions for domain adaptation to succeed in this setting (Ben-David & Urner, 2012; 2014) is the so-called weight ratio being lower bounded, which is equal to $\min_X d_h^{\pi, s}(X)/d_h^{\pi, t}(X)$ and $X$ is any measurable subset of the input space. That is exactly equal to the distributional shift term $C_h^f$ defined in the paper up to constant multipliers. Intuitively, there will be no information being transferred to the target in the worst case without such conditions.

