# OpenReview forum: "The Sample Complexity of Online Strategic Decision Making with Information Asymmetry and Knowledge Transportability"
_ICML.cc/2025/Conference — ICML 2025 poster_

### Official Review · Reviewer_Ju5y · 2025-03-09

**Overall Recommendation:** 2

**Summary:**

This paper studies online decision-making under information asymmetry and knowledge transportation. They formulate this problem using strategic MDP where an principle interacts with a sequence of myopic agents whose can impact the reward functions and transition kernels. The goal is for the principal to design a near-optimal policy that maximizes its total rewards when interacting with a target population of agents that might be different from the source population of agents during learning. The paper proposes a model-based algorithm that uses nonparametric instrumental variable method which can learn $\eps$-optimal policy using $O(1/\epsilon^2)$ samples.

**Claims And Evidence:**

I am completely unfamiliar with this problem setting, thus cannot judge the soundness of the theoretical claims in this paper.

**Essential References Not Discussed:**

The paper seems to discuss all relevant essential references to my best educated guess.

**Experimental Designs Or Analyses:**

This is a theory paper without any experimental results.

**Methods And Evaluation Criteria:**

I am completely unfamiliar with this problem setting, thus cannot judge the soundness of the proposed method in this paper.

**Other Comments Or Suggestions:**

I don't have any specific suggestions or comments.

**Other Strengths And Weaknesses:**

Strength:

- The paper seems to study an important problem of decision making in information asymmetry and information transportability.


Weaknesses:
- It's hard to follow the results in the paper, especially for those who are not familiar with the literature.
- I am not sure what technical novelty of the present paper compared to the algorithmic results and development in Yu et al 2022. I know the present paper considers an online setting instead of the offline setting of Yu et al 2022. Does generalize to the online setting require standard tools?
- To address transferability, the paper relies on an extremely stringent notion -- the worst-case density ratio -- worst case over all policy $\pi$ and $\nu$. First, this density ratio can be extremely large. In fact, most of the recent algorithmic developments in offline RL does not use the worst-case density ratio any more. Second, transferability under the worst-case density ratio looks trivial. Why would we consider transferability at all if the solution offered is trivial? Why don't focus on the core problems with meaningful results?

**Questions For Authors:**

Please address my questions in the weakness section.

**Relation To Broader Scientific Literature:**

The paper seems to discuss related works quite well by mentioning RL with confounded data, especially the discussion of Yu et al 2022 which considers a similar problem to the present paper yet in the offline setting instead of the online setting.

**Theoretical Claims:**

I am completely unfamiliar with this problem setting, thus cannot judge the correctness of any proofs for theoretical claims in this paper.

---

> ### Author Rebuttal · Authors · 2025-03-31
>
> We sincerely appreciate your valuable comments and suggestions!
>
> **Regarding the technical novelty compared to [1]**: In contrast to Yu et al. [1], this work studies the **online** strategic interaction model. Below, we briefly highlight the key technical novelties, with further details provided in Appendix B.3.
>
> 1. To better align with real-world scenarios, our online model explicitly incorporates knowledge transportability. A key motivation for this is that the agent population with which the principal interacts may evolve over time. For example, the personal attributes of job applicants for a company may vary across different periods. Thus, our model allows the agent type distribution in the online setting to differ from the target agent type distribution of the principal.
>
> 2. The online strategic interaction model **cannot be directly solved using standard online RL algorithms** due to endogenous noise (i.e., confounding variables) and the knowledge transportability challenge. This work introduces novel techniques to address these issues:
>
>    - While both this work and [1] leverage the NPIV model for estimating the empirical risk function $\hat{L}$, we construct a distinct confidence set with a different concentration analysis. Specifically, [1] derives a concentration bound under an i.i.d. dataset assumption, whereas our work handles non-i.i.d. data. Furthermore, [1] constructs the confidence set by bounding the value difference between a candidate model and the minimizer of $\hat{L}$, whereas we construct our confidence set solely by bounding the value of $\hat{L}$ for candidate models.
>
>    - We establish a key error propagation technique from the NPIV estimator’s error to the online regret using martingale concentration analysis with Freedman’s inequality.  Furthermore, we have developed cleaner proofs that rely **only** on the realizability assumption and the boundedness of zeros of a concave quadratic function, thereby simplifying the complex proof techniques in [1] that were based on the symmetric and star-shaped assumption.
>
> **Regarding the transferability assumption**: Knowledge transportability is considered for two main reasons. First, in many real-world applications, the principal interacts with a diverse or evolving agent population, making this assumption highly practical. Our algorithm can also accommodate scenarios where the source population $\mathcal{P}^s$ changes across different episodes. Second, if the source agent population were identical to the target agent population, standard online RL algorithms could be directly applied. However, when these populations differ, solving the problem becomes significantly more challenging.
>
> As for the "worst-case density ratio over all policies $\pi$ and distributions $\nu$", we note that this assumption can be generalized to an upper bound on the ratio of occupancy measures $$\frac{d^{\pi, s}_h}{d^{\pi, t}_h}$$
>
> for any policy $\pi$. Equivalently, it corresponds to bounding the density ratio of the agent type distribution, $\prod_{i \leq h} \mathcal{P}^s_i / \prod_{i \leq h} \mathcal{P}^t_i$. Ensuring this ratio remains bounded is crucial for addressing distributional shifts between the source and target populations. Our problem is analogous to the covariate shift setting in unsupervised domain adaptation, where the model uses $(s_h, a_h, e_h)$ to predict $(r_h, s_{h+1})$ based on underlying functions $R^*_h$ and $P^*_h$. The source distribution is $d^{\pi,s}_h$, and the target distribution is $d^{\pi,t}_h$
>
> for any policy $\pi$. A well-established condition for successful domain adaptation [2,3] requires a lower-bounded weight ratio, defined as $\min_{X} d^{\pi,s}_h(X) / d^{\pi,t}_h(X)$ for any measurable subset $X$ of the input space. This precisely corresponds to the distribution shift term $C^f_h$ in our paper, up to constant multipliers. Intuitively, without such conditions, knowledge transfer to the target domain would be infeasible.
>
> We hope our rebuttal has clarified the reviewer’s confusion and respectfully hope that the reviewer would consider re-evaluating the merit of our work accordingly.
>
> [1]. Yu, Mengxin, Zhuoran Yang, and Jianqing Fan. "Strategic decision-making in the presence of information asymmetry: Provably efficient rl with algorithmic instruments." arXiv preprint arXiv:2208.11040 (2022).
>
> [2]. Ben-David, Shai, and Ruth Urner. "On the hardness of domain adaptation and the utility of unlabeled target samples." International Conference on Algorithmic Learning Theory. Berlin, Heidelberg: Springer Berlin Heidelberg, 2012.
>
> [3]. Ben-David, Shai, and Ruth Urner. "Domain adaptation–can quantity compensate for quality?." Annals of Mathematics and Artificial Intelligence 70 (2014): 185-202.

---

### Official Review · Reviewer_6m6G · 2025-03-12

**Overall Recommendation:** 3

**Summary:**

This work considers a principal-agent RL framework, where the reward and transitions also depend on the unobserved action.

In this framework, the authors propose an algorithm that will learn the principal rewards, but also --- and this is the challenging part because of the fact that only partial, cofounding observations are available on the agent action --- the agent reward and transition probabilities. For that they use knowledge transfer methods, and derive a sample complexity upper bound for their algorithm.

**Claims And Evidence:**

see below

**Essential References Not Discussed:**

see below

**Experimental Designs Or Analyses:**

see below

**Methods And Evaluation Criteria:**

see below

**Other Comments Or Suggestions:**

My main concern with this work is that it is based on the claim that the considered framework "is more challenging than classic RL". I however disagree with this claim, as we could still apply any RL algorithm to that framework, ignoring the agent feedback and intervention.
Indeed, the principal could just learn using the marginal rewards $\bar{R}^*$ and transitions $\bar{P}^*$.

$e_h$ seems to only correspond to additional feedback (wrt to classical RL framework), that can enhance learning, through a learning of the agent reward functions (and transition). But then, I would like more convincing results towards the fact that the proposed algorithm and derived bounds yield some improvement wrt typical RL algorithms. Notably, the sample complexity bound of Theorem 5.4 seems to be similar to the typical one in RL, if we omit the dependency in the number of states. Here is the number of states does not appear, but might be hidden into new terms such as the $\mathcal{R}$ or $\mathcal{P}$.

As a consequence, I would like to have a concrete example by the authors (e.g., if the reward class for the agent is linear) that clearly yields an improvement in terms of sample complexity with respect to typical RL bounds. Additionally, some experiments might be helpful.

Currently, I feel that the algorithm might learn unnecessary things, such as $R^a$ and $P$, while learning $\bar{R}^*$ and $\bar{P}^*$ should be much simpler and competitive.

**Other Strengths And Weaknesses:**

see below

**Questions For Authors:**

- Why couldn't we apply typical RL algorithms to your framework, with the suggestion made above?

**Relation To Broader Scientific Literature:**

see below

**Theoretical Claims:**

see below

---

> ### Author Rebuttal · Authors · 2025-03-31
>
> We sincerely appreciate your valuable comments and suggestions!
>
> **Regarding the limitations of standard online RL algorithms in our framework**: The core reason standard online RL algorithms cannot be directly applied is that the online strategic interaction model under the source agent distribution (denoted by $\mathcal{M}^*(\mathcal{P}^s)$) differs from the one under the target agent distribution (denoted by $\mathcal{M}^*(\mathcal{P}^t)$). Simply ignoring feedback will lead to model misspecification and linear regret, as the existence of confounders leaves you a wrong distribution when calculating $\bar R^*$ and $\bar P^*$. Below, we provide a detailed explanation of this distinction and why it arises.
>
> The knowledge transportability framework in this paper requires the principal to explore the optimal policy of an online strategic interaction model when interacting with a different agent type distribution than the one used for online data collection. Since the model under the source agent distribution $\mathcal{M}^*(\mathcal{P}^s)$ (which can also be viewed as an MDP per Section 3.1 of the paper) **is not the same as** the model under the target agent distribution $\mathcal{M}^*(\mathcal{P}^t)$, standard online RL algorithms cannot be used to find the optimal policy for $\mathcal{M}^*(\mathcal{P}^t)$.
>
> A key motivation for studying knowledge transportability is that the agent population with which the principal interacts may change over time. For example, the personal attributes of job applicants at a company can vary across different periods. To capture such scenarios, our model allows the agent type distribution during the online interactions to differ from the target. We also provide Example 3.2 in the paper as a motivating illustration.
>
> If the source and target agent populations were **identical**, standard online RL algorithms could be directly applied. However, when the populations differ, solving the problem becomes considerably more challenging.
>
> **Regarding the experiments**:
>
> We conducted a small-scale experiment in the tabular setting, motivated by Example 3.1. Consider a company (the principal) is recruiting project managers (PMs, the agents), and the company needs to determine the salary of the PMs.
>
> A corresponding bandit setting is constructed as follows: There is a single state and $H=1$, with two candidate actions, $H$ (high salary) and $L$ (low salary). The agent’s private type $t$ can be either diligent ($d$) or lazy ($l$). The principal’s feedback, representing project performance, can be good ($G$) or bad ($B$). The reward function $R^*(a, e)$ takes as input the action $a$ and feedback $e$. The feedback distribution $F(a, t)$ is defined as follows:
> - If $a=H$ and $t=d$, the feedback is always $G$.
> - If $a=L$ and $t=l$, the feedback is always $B$.
> - Otherwise, feedback is equally likely to be $G$ or $B$.
>
> In our experiment, we set $R^*(H, G) = 1.5, R^*(H, B) = 0, R^*(L, G) = 2, R^*(L, B) = 1$. The source agent distribution is $0.4$ for $d$ and $0.6$ for $l$, while the target distribution is $0.8$ for $d$ and $0.2$ for $l$. The principal's optimal action is $H$ for the target distribution and $L$ for the source distribution.
>
> The empirical risk function $\hat{L}^k$ in episode $k$ has the closed-form:
> $$\hat{L}^k(R) = \max_{f=(f_H, f_L)} \sum_{\tau=1}^k \left(f(a_\tau) (R(a_\tau, e_\tau) - r_\tau) - \frac{f^2(a_\tau)}{2} \right).
> $$
> Taking $f_H$ as an example, the closed-form solution is $\sum_{a_\tau = H} R(a_\tau, e_\tau) - r_\tau$. The solution for $f_L$ follows similarly. To simplify, we discretize the entries of $R(a, e)$ so that each takes values from \{0, 0.5, 1, 1.5, 2\}, leading to $5^4 = 625$ candidate models initially.
>
> The following table summarizes the results, illustrating the convergence of our algorithm. We ran experiments with three random seeds, using $\beta = 600$ for 1800 episodes. In all cases, $R^*$ was successfully recovered. The reported values reflect all three seeds.
>
> | Episode    | 300 | 600 | 900 | 1200 | 1500 | 1800 |
> | -------- | ------- | -------- | ------- | -------- | ------- | ------- |
> | Remaining models (625 in total)  | 25,32,37  | 5,14,14 | 1,5,2 | 1,2,2 | 1,2,1 | 1,1,1 |
> | True action $H$ taken (percentage) |  29.7%,31.3%,27.3%  | 14.8%,15.7%,22.2% | 43.2%,16.8%,34.8% | 57.4%,37.6%,51.1% | 65.9%,50.1%,60.1% | 71.6%,53.1%,67.4% |
>
> We hope our rebuttal has clarified the reviewer’s confusion and respectfully hope that the reviewer would consider re-evaluating the merit of our work accordingly.

---

> > ### Comment · Reviewer_6m6G · 2025-04-03
> >
> > I thank the authors for their answer. I now understand better the model at hand, and find it very interesting. I thus decide to raise my score in consequence. I would however recommend the authors to emphasize more on this source to target generalization setup when mathematically introducing the problem, as it was obviously unclear to me while reading the paper. Additionally, a nice presentation of these experiments (e.g. adding a comparison with typical RL baselines) would help in motivating the considered problem/method
> >
> > If I understand well, this would mean that the learner knows in advance the target distribution $\mathcal{P}^t$, while having no knowledge of the source one $\mathcal{P}^s$. How is that a reasonable assumption in the typical applications mentioned in the paper?

---

> > > ### Author Response · Authors · 2025-04-05
> > >
> > > We thank the reviewer for re-evaluating the paper and providing useful suggestions on the presentation of the formulation and experiments of the paper.
> > >
> > > **Regarding why assuming the principal knows $\mathcal{P}^t$ but does not know $\mathcal{P}^s$**: We address this by discussing both the underlying motivation (reflecting typical real-world scenarios) and the technical analysis.
> > >
> > > In terms of the motivation, the discrepancy between $\mathcal{P}^s$ and $\mathcal{P}^t$ mirrors common economic events. It is crucial that the principal has some preliminary observations of the target population (e.g., through early surveys), as the quality of these observations greatly influences the effectiveness of the principal’s policy. Without any knowledge of the target population, finding an optimal policy would be impossible. However, actively interacting with the target population is often challenging. For instance, a company may aim to recruit employees from a specific group but might only receive applications from the broader society, and our model is able to solve this problem as long as the target population can be approximated. Similarly, consider a scenario where a new medical treatment is tested in Country A, yet the government is interested in its effects in Country B; conducting the treatment in Country B may be impractical but obtaining demographics can be easier. We have also provided some discussions in the Introduction and Appendix C.2 of the paper.
> > >
> > > From a technical standpoint, the principal's optimal policy is inherently dependent on the target distribution, $\mathcal{P}^t$. Consequently, possessing knowledge of $\mathcal{P}^t$—or at least a reasonable approximation—is essential to determining the optimal policy. This assumption is standard in related fields, such as Myerson’s auction theory and the coordination theory of principal-agent problems [1, 2].
> > >
> > > [1]. Myerson, Roger B. "Optimal auction design." Mathematics of operations research 6.1 (1981): 58-73.
> > >
> > > [2]. Myerson, Roger B. "Optimal coordination mechanisms in generalized principal–agent problems." Journal of mathematical economics 10.1 (1982): 67-81.
> > >
> > > Thanks for your comments! We'll include this discussion in the camera-ready version.

---

### Official Review · Reviewer_o5zg · 2025-03-14

**Overall Recommendation:** 3

**Summary:**

This submission investigates online strategic decision-making in multi-agent environments characterized by information asymmetry and knowledge transportability. Specifically, it addresses the challenge of learning optimal decision policies when agents have private information that introduces confounding factors, and when direct experimentation in the target environment is infeasible, thus requiring knowledge transfer from another, easier-to-study domain. To tackle these issues, the authors propose an online strategic interaction model and employ a nonparametric instrumental variable (NPIV) approach for causal identification to handle confounding. Coupled with optimistic planning, their algorithm effectively transfers learned causal insights between different populations. Theoretically, the submission shows that the proposed approach achieves near-optimal policy learning with a tight sample complexity, explicitly characterizing how information asymmetry and differences between source and target domains impact learning efficiency.

## update after rebuttal

I read the authors' rebuttal. Although there is a slight misunderstanding in their interpretation of the cited paper, I generally agree with their revisions. I tend to maintain the current score.

**Claims And Evidence:**

Clear.

**Essential References Not Discussed:**

Mentioned before.
- Bernasconi, M., Castiglioni, M., Marchesi, A., & Mutti, M. (2023). Persuading farsighted receivers in mdps: the power of honesty. _Advances in Neural Information Processing Systems_, _36_, 14987-15014.
- Lin, Y., Li, W., Zha, H., & Wang, B. (2023). Information design in multi-agent reinforcement learning. _Advances in Neural Information Processing Systems_, _36_, 25584-25597.

**Experimental Designs Or Analyses:**

No.

**Methods And Evaluation Criteria:**

#### Issue 1
Line 123.
> We consider the time-inhomogeneous Markov policy class $\Pi$ in this work.

Why do you concern Markov policies rather than history-dependent policies? It would be helpful if the authors elaborate the reason. Sometimes history-dependent policies are necessary. For example, see the discussion in Section 3 of this paper:

- Bernasconi, M., Castiglioni, M., Marchesi, A., & Mutti, M. (2023). Persuading farsighted receivers in mdps: the power of honesty. _Advances in Neural Information Processing Systems_, _36_, 14987-15014.

#### Issue 2

Figure 1 illustrates the timeline of their proposed model. This model specifies the focused problem. But it is very similar to the Markov signaling game proposed in the following paper:
- Lin, Y., Li, W., Zha, H., & Wang, B. (2023). Information design in multi-agent reinforcement learning. _Advances in Neural Information Processing Systems_, _36_, 25584-25597.
Especially the extensions mentioned in it. So it would be helpful if the authors provide the comparison between them.

**Other Comments Or Suggestions:**

- "casual" should be "causal" in several places.
- "confounded" is a bad word. I suppose it should be "confounding".
- Figure 1 and 2 are too big.

**Other Strengths And Weaknesses:**

Provided examples in Section 3 are helpful.

**Questions For Authors:**

Mentioned before.

**Relation To Broader Scientific Literature:**

The author's discussion in the related work section is relatively comprehensive.

**Theoretical Claims:**

No.

This is an emergency review task for me. I only saw the invitation two hours before the deadline, so it was impossible for me to review it thoroughly.

---

> ### Author Rebuttal · Authors · 2025-04-01
>
> We appreciate your insightful comments and feedback!
>
> **Regarding the Markov policy class**: We focus on the Markov policy class because any online strategic interaction model has an **optimal Markov policy**. This follows from the fact that an online strategic interaction model is equivalent to an MDP when the agent's private type distribution is given. Please refer to Section 3.1 of the paper for further details.
>
> **Regarding the comparison with [1]**: While [1] also investigates a multi-agent generalization of the principal-agent problem, our work differs in formulation, methodology, and analysis.
>
> - **Formulation**: [1] adheres to the traditional principal-agent framework, where the principal designs an incentive-compatible signaling scheme, and the agent responds based on the state and the principal's generated signal. In their setting, the principal has an information advantage over the agent, and their rewards depend on the global state and the agent’s observable action. In contrast, our model fundamentally differs: the agent possesses private information hidden from the principal, and the agent’s unobservable private actions directly influence the principal’s reward. Additionally, both the principal and agents independently maximize their own utility functions without the incentive-compatible constraint. As a result, the decision-making processes in our model and theirs follow different logical structures.
>
> - **Methodology and Analysis**: We propose a provably sample-efficient algorithm leveraging the NPIV method and optimistic planning. Our work includes a rigorous statistical analysis establishing the sample complexity of our algorithm. In contrast, [1] adopts a policy gradient approach and provides extensive empirical evidence to demonstrate its effectiveness.
>
> **Regarding the typos**: Thank you for point out them, and we will fix them accordingly.
>
> [1]. Lin, Yue, et al. "Information design in multi-agent reinforcement learning." Advances in Neural Information Processing Systems 36 (2023): 25584-25597.

---

### Official Review · Reviewer_Whc9 · 2025-03-15

**Overall Recommendation:** 3

**Summary:**

The authors consider a principal-agent problem where the principal interacts with a sequence of strategic agents with private types drawn from a different distribution than one the principal has information about. The principal's reward depends on unobserved confounders, so the authors propose using an instrumental variable technique to faithfully estimate quantities of interest, before applying variants of the sample complexity analysis for model-based RL.

## Update After Rebuttal
The authors added in a discussion of some of the points of confusion I originally had to the paper. I already factored this into my evaluation of the paper, and hence maintain my score

**Claims And Evidence:**

Yes.

**Essential References Not Discussed:**

Could you add in a reference to https://arxiv.org/abs/2202.01312? I also think Chen & Pouzo '12 is the right citation for the measure of ill-posedness you consider in the paper (https://eml.berkeley.edu/~dpouzo/pdfs/cp-rate-webpage-jan-11.pdf).

**Experimental Designs Or Analyses:**

There are no experiments, unfortunately.

**Methods And Evaluation Criteria:**

No experimental methods.

**Other Comments Or Suggestions:**

- Could you match the colors of the variables in Figure 1 and Figure 2? Also, could you use different colors for Figure 2 (i.e. having a separate color for IVs vs confounders) and add in $\xi$ to the SCM?

- I'd suggest cutting / reworking Example 3.2 -- it seems entirely disconnected from the paper.

- I'd suggest adding in a simple experiment on a tabular problem -- I think this should be fairly easy to do as you could compute the discriminators / Lagrange multipliers in closed form.

- I think there's a typo on lines 259-260 re: where the word "principle" appears in the sentence.

- I think it should be "single-policy" concentrability in Defn. 5.3 -- do you mind checking this?

- For the specific case of the game-theoretic IV algorithms, it should be fairly easy to add in computational efficiency results via the standard no-regret machinery. It might be interesting to do so to complement your statistical results.

**Other Strengths And Weaknesses:**

At some level, this paper is a combination of several fairly well explored techniques (MBRL, instrumental variables, transportability). I think it is technically interesting to be able to combine these, but I'm not quite sure how useful / impactful this combination will be,

**Questions For Authors:**

1) A common critique of instrumental variable methods is that the zero mean, additive confounding assumption is unreasonable for a lot of applications. Could you comment on specific scenarios where this might be approximately true and add them to the paper?

2) Why did you choose the minimax estimators for IV rather than either the "DeepIV" / generative modeling approaches (https://proceedings.mlr.press/v70/hartford17a/hartford17a.pdf) or the DFIV techniques (https://openreview.net/pdf?id=sy4Kg_ZQmS7)? I don't think the game-theoretic formulation is fundamental to your claims here so you might be able to instead frame your paper as a framework rather than analysis of a particular algorithm.

3) You assume realizability for the discriminators your training. Loosely speaking, I think of these as being Lagrange multipliers for the vector of conditional moment restrictions. Lagrange multipliers can often need to take unbounded values to actually enforce constraints are satisfied. Could you discuss why a finite bound $B$ is a reasonable assumption?

**Relation To Broader Scientific Literature:**

Essentially, this paper does analysis for MBRL with confounders / transportability.

**Theoretical Claims:**

I read all the theorems and skimmed the proofs and nothing struck me as clearly false.

---

> ### Author Rebuttal · Authors · 2025-04-01
>
> Thanks for your useful comments and suggestions!
>
> **Regarding the minimax estimators for IV**:
>
> We employ F-R duality to formulate a minimax estimator instead of using DeepIV or DFIV for the following reasons:
>
> 1. Our primary objective is to design **provably sample-efficient exploration algorithms** for real-world multi-agent decision-making systems with information asymmetry and knowledge transportability. The minimax NPIV method serves as a tool to address confounding issues in our framework. Additionally, analyzing its statistical efficiency is crucial for proving the sample complexity of our exploration algorithm. While the minimax approach aligns well with our analysis, DeepIV and DFIV rely on deep neural networks, making statistical analysis challenging.
>
> 2. DeepIV and DFIV follow a 2SLS framework for IV regression in linear settings. Our framework generalizes beyond the linear case, as demonstrated in Section 5.2, where we show how our setting reduces to the linear case. Consequently, a more general algorithmic framework is necessary.
>
> **Regarding the bounded Lagrangian multipliers**: The Lagrangian multipliers can be bounded because the **concentration analysis of the NPIV model does not require them to take optimal values**. We discuss the relevant bounds in Lemma H.4 (Appendix H), where we show that the desired values of the Lagrangian multipliers for the concentration analysis remain within a constant range.
>
> **Regarding the zero mean of the instrumental variable**: We will expand Example 3.1 to illustrate a specific scenario where the zero mean assumption holds. Consider a company recruiting project managers (PMs) to lead projects. The company determines PM salary levels, with states corresponding to the company's status (e.g., stock price) and actions representing salary decisions by the board. The PMs' private types can be diligent or lazy. The zero mean assumption requires the company’s reward to be approximated by an underlying reward function $R^*$ under any agent type distribution. This is reasonable because the feedback $e_h$ is able to capture the bias introduced by the agent type distribution. For example, when there are more diligent PMs, the company receives more positive feedback, leading to higher rewards with the same $R^*$ but a higher probability of positive feedback. Generally speaking, we can "move" the mean into the reward $R^*$ through the agent type distribution. Notably, the zero mean assumption does not require the noise to be zero mean given a specific agent type.
>
> **Regarding the experiments**:
>
>  We conducted a small-scale experiment in the tabular setting, motivated by Example 3.1.
>
> A corresponding bandit setting is constructed as follows: There is a single state and $H=1$, with two candidate actions, $H$ (high salary) and $L$ (low salary). The agent’s private type $t$ can be either diligent ($d$) or lazy ($l$). The principal’s feedback, representing project performance, can be good ($G$) or bad ($B$). The reward function $R^*(a, e)$ takes as input the action $a$ and feedback $e$. The feedback distribution $F(a, t)$ is defined as follows:
> - If $a=H$ and $t=d$, the feedback is always $G$.
> - If $a=L$ and $t=l$, the feedback is always $B$.
> - Otherwise, feedback is equally likely to be $G$ or $B$.
>
> In our experiment, we set $R^*(H, G) = 1.5, R^*(H, B) = 0, R^*(L, G) = 2, R^*(L, B) = 1$. The source agent distribution is $0.4$ for $d$ and $0.6$ for $l$, while the target distribution is $0.8$ for $d$ and $0.2$ for $l$. The principal's optimal action is $H$ for the target distribution and $L$ for the source distribution.
>
> The empirical risk function $\hat{L}^k$ in episode $k$ has the closed-form:
> $$\hat{L}^k(R) = \max_{f=(f_H, f_L)} \sum_{\tau=1}^k \left(f(a_\tau) (R(a_\tau, e_\tau) - r_\tau) - \frac{f^2(a_\tau)}{2} \right).
> $$
> Taking $f_H$ as an example, the closed-form solution is $\sum_{a_\tau = H} R(a_\tau, e_\tau) - r_\tau$. The solution for $f_L$ follows similarly. To simplify, we discretize the entries of $R(a, e)$ so that each takes values from \{0, 0.5, 1, 1.5, 2\}, leading to $5^4 = 625$ candidate models initially.
>
> The following table summarizes the results, illustrating the convergence of our algorithm. We ran experiments with three random seeds, using $\beta = 600$ for 1800 episodes. In all cases, $R^*$ was successfully recovered. The reported values reflect all three seeds.
>
> | Episode    | 300 | 600 | 900 | 1200 | 1500 | 1800 |
> | -------- | ------- | -------- | ------- | -------- | ------- | ------- |
> | Remaining models (625 in total)  | 25,32,37  | 5,14,14 | 1,5,2 | 1,2,2 | 1,2,1 | 1,1,1 |
> | True action $H$ taken (percentage) |  29.7%,31.3%,27.3%  | 14.8%,15.7%,22.2% | 43.2%,16.8%,34.8% | 57.4%,37.6%,51.1% | 65.9%,50.1%,60.1% | 71.6%,53.1%,67.4% |
>
> **Regarding the presentations and references**: Thank you for your suggestions. We will revise the presentation and incorporate the recommended references accordingly.

---

### Decision · Program_Chairs · 2025-05-01

**Decision:**

Accept (poster)

**Comment:**

We had a fruitful discussion about this paper. Overall the sentiment was rather positive (3 weak accepts, and 1 weak reject from a reviewer which is less familiar with the topic). I think our general assessment is well summarized by Reviewer Whc9, who writes "At some level, this paper is a combination of several fairly well explored techniques (MBRL, instrumental variables, transportability). I think it is technically interesting to be able to combine these, [but it also not ground breaking]".

Nitbit: The paper is partially motivated by the generalized principal-agent problem (Myerson 1982), and explicitly mentions "contract design" as a motivation, but fails to cite any of the rapidly emerging literature on "algorithmic contract theory" (esp. on sample complexity and typed agent settings).